# NOT-SO-BIG-GAN: GENERATING HIGH-FIDELITY IMAGES ON SMALL COMPUTE WITH WAVELET-BASED SUPER-RESOLUTION

## ABSTRACT

State-of-the-art models for high-resolution image generation, such as BigGAN and VQVAE-2, require an incredible amount of compute resources and/or time (512 TPU-v3 cores) to train, putting them out of reach for the larger research community. On the other hand, GAN-based image super-resolution models, such as ESRGAN, can not only upscale images to high dimensions, but also are efficient to train. In this paper, we present NOT-SO-BIG-GAN (NSB-GAN), a simple yet cost-effective two-step training framework for deep generative models (DGMs) of high-dimensional natural images. First, we generate images in low-frequency bands by training a *sampler* in the wavelet domain. Then, we super-resolve these images from the wavelet domain back to the pixel-space with our novel wavelet super-resolution *decoder* network. Wavelet-based down-sampling method preserves more structural information than pixel-based methods, leading to significantly better generative quality of the low-resolution sampler (e.g., 64×64). Since the sampler and decoder can be trained in parallel and operate on much lower dimensional spaces than end-to-end models, the training cost is substantially reduced. On ImageNet 512×512, our model achieves a Fréchet Inception Distance (FID) of 10.59 – beating the baseline BigGAN model – at half the compute (256 TPU-v3 cores).

## 1 INTRODUCTION

Generative modeling of natural images has achieved great success in recent years (Kingma & Welling, 2013; Goodfellow et al., 2014; Arjovsky et al., 2017; Menick & Kalchbrenner, 2019; Zhang et al., 2018a). Advancements in scalable computing and theoretical understanding of generative models (Miyato et al., 2018; Zhang et al., 2018a; Gulrajani et al., 2017; Mescheder et al., 2018; 2017; Roth et al., 2017; Nowozin et al., 2016; Srivastava et al., 2017; 2020; Karras et al., 2020), have, for the first time, enabled the state-of-the-art techniques to generate photo-realistic images in higher dimensions than ever before (Brock et al., 2018; Razavi et al., 2019; Karras et al., 2020). Yet, generating high-dimensional complex data, such as ImageNet, still remains challenging and extremely resource intensive. At the forefront of high-resolution image generation is BigGAN (Brock et al., 2018), a generative adversarial network (GAN) (Goodfellow et al., 2014) that tackles the curse of dimensionality (CoD) head-on, using the latest in scalable GPU-computing. This allows for training BigGAN with large mini-batch sizes (e.g., 2048), which greatly helps to model highly diverse, large-scale datasets like ImageNet. But, BigGAN's ability to scale to high-dimensional data comes at the cost of a hefty compute budget. A standard BigGAN model at 256×256 resolution can require up to a month or more of training time on as many as eight Tesla V100 graphics processing units (GPUs). This compute requirement raises the barrier to entry for using and improving upon these technologies as the wider research community may not have access to any specialized hardware (e.g., Tensor processing units (TPUs) (Jouppi et al., 2017). The environmental impact of training large-scale models can also be substantial as training BigGAN on 512×512 images with 512 TPU cores for two days reportedly used as much electricity as the average American household does in about six months (Schwab, 2018).

Motivated by these problems, we present NOT-SO-BIG-GAN (NSB-GAN), a small compute training alternative to BigGAN, for class-conditional modeling of high-resolution images. In end-to-end

generative models of high-dimensional data, such as VQVAE-2 (Razavi et al., 2019) and Karras et al. (2017), the lower layers transform noise into low resolution images, which are subsequently upscaled i.e. *super-resolved* to higher dimensions in the higher layers. Based on this insight, in NSB-GAN we propose to split the end-to-end generative model into two separate neural networks, a sampler and an up-scaling decoder that can be trained in parallel on much smaller dimensional spaces. In turn, we drastically reduce the compute budget of training. This split allows the sampler to be trained in up to 16-times lower dimensional space, not only making it compute efficient, but also alleviating the training instability of end-to-end approaches. To this end, we propose wavelet-space training of GANs. As compared to pixel-based interpolation methods for down-sampling images, wavelet-transform (WT) (Haar, 1909; Daubechies, 1992; Antonini et al., 1992) based down-sampling preserves much more structural information, leading to much better samplers in fairly low resolutions (Sekar et al., 2014). When applied to a 2D image, wavelet transform slices the image into four equally-sized image-like patches along different frequency bands. This process can be recursively applied multiple times in order to slice a large image into multiple smaller images, each representing the entire image in different bandwidths. This is diagrammatically shown in Figure 1. Here, the top-left patch (TL) lies in the lowest frequency band and contains most of the structure of the original image and therefore the only patch preserved during downsampling. The highly sparse top-right (TR), bottom-left (BL) and bottom-right (BL) patches lie in higher bands of frequency and are therefore dropped. But wavelet-space sampling prohibits the use of existing pixel-space super-resolution models, such as Ledig et al. (2017); Wang et al. (2018), to upscale the samples. Thus, we introduce two wavelet-space super-resolution *decoder* networks that can work directly with wavelet-space image encoding, while matching the performance of equivalent pixel-space methods. Training our decoders is extremely compute efficient (e.g., 3 days on the full ImageNet dataset), and, once trained on a diverse dataset like ImageNet, can generalize beyond the original training resolution and dataset.

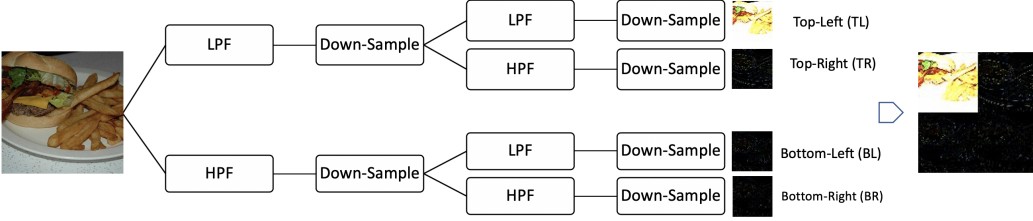

Figure 1: Wavelet transformation consists of a low-pass and a high-pass filter, followed by a down-sampling step that splits the image into two equal-sized patches. Each of these two patches undergo the same operation again resulting in four equal-sized patches, TL, TR, BL and BR.

Our main contributions are the following:

- We introduce a simple training framework NSB-GAN for large generative models and show that super-resolving low-resolution samplers reduces the cost of training by orders of magnitude without sacrificing image quality and even outperforms the baseline BigGAN model at higher resolution (e.g., $512 \times 512$).
- In addition to a simple pixel-based version of NSB-GAN, we introduce wavelet-space training of DGMs and demonstrate that it leads to better image quality compared to pixel-space NSB-GAN model on average.
- While conceptually simple (especially in the pixel-space), training large models to work in tandem requires careful engineering. Therefore, we make our entire NSB-GAN code-base publicly available to the wider research community.

## 2 BACKGROUND AND RELATED WORK

Given a set $X = \{x_i | \forall i \in \{1, \ldots N\}, x_i \in \mathbb{R}^D\}$ of samples from the true data distribution $p(x)$, the task of deep generative modeling is to approximate this distribution using deep neural networks. All generative models of natural images assume that $p(x)$ is supported on a $K$-dimensional, smaller

manifold of $\mathbb{R}^D$. As such, they model the data using a conditional distribution $p_\theta(x|z)$, where $\theta$ represents the model parameters and $Z \in \mathbb{R}^K$ is the latent variable with the prior $p(z)$. $Z$ is marginalized to obtain the log-likelihood (LLH) $\log p_\theta(x) = \log \int p(z)p_\theta(x|z)dz$ of $X$ under the model. Generative model are trained by either maximizing this LLH or its lower-bound. There are two types of deep generative models, explicit models like variational autoencoders (VAE) (Kingma & Welling, 2013) that have an explicit form for the LLH and implicit models such as GANs (Goodfellow et al., 2014) that do not have such a form. VAEs use a decoder network $D_\theta$ to parameterize the model distribution $p_\theta(x|z)$ and an encoder network $E_\phi$ to parameterize a variational distribution $q_\phi(z|x)$ as an approximation to the true posterior under the model. They can then be trained using the evidence lower-bound (ELBO), $\log p_\theta(x) \geq -\text{KL}[q_\phi(z|x)\|p(z)] + \int q_\phi(z|x)\log p_\theta(x|z)dz$. KL here refers to the Kullback–Leibler divergence between $q_\phi(z|x)$ and $p(z)$. Unlike VAEs, GANs do not make any assumptions about the functional form of the conditional distribution $p_\theta(x|z)$ and, therefore, do not have a tractable likelihood function. Instead, they directly attempt to minimize either a variational lowerbound to a chosen $f$-divergence (Nowozin et al., 2016; Srivastava et al., 2017) or integral probability metric (IPM) (Arjovsky et al., 2017; Li et al., 2017; Srivastava et al., 2020) between the model distribution $p_\theta(x)$ and $p(x)$ using binary classifier based density ratio estimators.

Though the idea of multiscale modeling in GANs has been considered throughly in Denton et al. (2015); Karras et al. (2017; 2020); Liu et al. (2019b), our method is more closely related to VQVAE-2 (Razavi et al., 2019) and the Subscale Pixel Network (SPN) (Menick & Kalchbrenner, 2019) models. VQVAE-2 is a hierarchical version of the VQVAE model (van den Oord et al., 2017) that unlike VAEs, does not impose a KL-based regularization on the latent space. Instead, VQVAE/VQVAE2 models use vector quantization as a strong functional prior. As such, sampling from the latent space is not straightforward and requires a large compute budget to train auto-regressive PixelSNAIL (Chen et al., 2017) models to estimate the density on the latent space to generate samples. (Menick & Kalchbrenner, 2019) shows that it is easier to generate high-resolution images by splitting the process into two steps. In step one, they slice the image into small patches and then learn to generate these patches using a patch-level autoregressive model. This step allows for capturing the general structure of the image but misses the finer details. Therefore, in step two, they train another network that learns to fill in the missing details. The SPN model has a significant shortcoming, however, as its patch-level autoregressive approach fails to capture long term dependencies (e.g. eyes on a face looking in different directions). Similar to SPN's multi-scale approach, NSB-GAN decouples structure learning from adding details. Unlike SPN, however, it does not suffer from the long-term dependency problem. This is because NSB-GAN uses wavelet transformation to slice the image, which naturally decouples the structure (TL) from the details (TR, BL, BR) in the image. We demonstrate this difference diagrammatically in the Appendix F. More importantly, wavelet transform alleviates the need for autoregressive modeling of the patches as each patch represents the entire image in different frequency bands.

**Image Super Resolution**   Since the pioneering work of SRCNN (Dong et al., 2014) to tackle the problem of single image super-resolution, recovering a high-resolution image from a low-resolution image, with a deep neural network, deep convolutional neural network approaches have demonstrated great success reconstructing realistic high-resolution images. ESRGAN (Wang et al., 2018) is the state-of-the-art GAN model that majorly builds on SRGAN, RDN, and EDSR (Ledig et al., 2017; Zhang et al., 2018b; Lim et al., 2017). Improving upon its predecessor SRGAN, it modifies the SRGAN's architecture from SRResNet to Residual-in-Residual Dense Block (RRDB) without batch normalization, finetunes the perceptual loss implemented with the VGG-19 model (Simonyan & Zisserman, 2015), and utilizes a relativistic GAN-like (Jolicoeur-Martineau, 2019) adversarial loss to predict relative realness. First, only the generator is trained as a Peak Signal-to-Noise Ratio (PSNR)-oriented model with L1 loss. Then, the GAN is trained as a whole, initialized from this pre-training, with carefully balanced L1, perceptual, and adversarial losses. With such, ESRGAN is able to reconstruct realistic high-frequency texture and details missing in the low-resolution images.

## 3   METHOD

NSB-GAN training introduces a fairly simple change to the original BigGAN model. Instead of training the generator directly on the full dimensionality of the data, $D = 256$ (say, for ImageNet dataset), we train the generator in $K = 64$. Then, in order to upscale the $64 \times 64$ image back

to 256, we train an up-scaling decoder. We now describe two specific instances of the NSB-GAN framework, NSB-GAN-W and NSB-GAN-P in detail.

## 3.1 NSB-GAN-W

NSB-GAN-W is best described as an autoencoder that operates in the frequency domain instead of the usual pixel space. It is comprised of a deterministic encoder, a learned decoder and learned prior network, i.e. sampler. These three components form a full generative model that can produce high-quality, high-resolution samples with a week of training on commodity hardware.

### 3.1.1 DETERMINISTIC ENCODER.

The NSB-GAN-W encoder is a deterministic function that recursively applies wavelet transform to the input image, *retaining only the TL patches at each level*. Each TL patch is a quarter of the size of the previous TL patch, resulting in a pyramid-like stack of 2D patches after multiple transformations. The last TL contains the lowest frequencies and, therefore, the most structural information about the image. We use it as a compressed, lossy representation of the input image in our encoding step.

For a $N^2$-dimensional image $X$, let us denote the matrix obtained after wavelet transform as $W(X)$, which is a $N \times N$ block matrix with the following structure,

$$W(X) = \left[ \begin{array}{c|c} W_{1,1}(X) & W_{1,2}(X) \\ \hline W_{2,1}(X) & W_{2,2}(X) \end{array} \right].$$

Here, with a slight abuse of notation, $W_{1,1}(X)$ represents the $\left(\frac{N}{2} \times \frac{N}{2}\right)$-dimensional TL patch of image X. NSB-GAN's encoder $\mathcal{E}$ can then be defined recursively as follows,

$$\begin{aligned} \mathcal{E}_0(X) &= X \\ \mathcal{E}_l(X) &= W_{1,1}(\mathcal{E}_{l-1}(X)), \quad 1 \le l \le L \end{aligned} \tag{1}$$

where $l$ is the number of wavelet transforms applied to the input image. As can be seen, the size of the retained TL patch decreases exponentially with $l$.

### 3.1.2 DECODER

After wavelet transform, the original image can be deterministically recovered from the TL, TR, BL, and BR patches using IWT. NSB-GAN-W, however, discards the high-frequency patches during its encoding step, rendering a fully deterministic decoding impossible. Thus, the NSB-GAN decoder first learns to recover the missing high-frequency patches (using a neural network) and then deterministically combines them using IWT to reconstruct the original input. We now present two NSB-GAN-W decoder designs based on ResNet (He et al., 2016; Ledig et al., 2017) and UNet (Ronneberger et al., 2015) architectures respectively.

**ESRGAN-W** Since NSB-GAN-W sampler generates samples in the wavelet domain, we modify the basic ESRGAN architecture (Wang et al., 2018) to allow it to accept images in the wavelet-domain, up-scale them, and then project them back to the pixel-space. Specifically, we replace the generator in ESRGAN with a 16-block-deep SRResNet without batch normalization (Ledig et al., 2017), denoted by $f_\theta$, and replaced the bilinear interpolation based up-scaling with IWT. We can then define the decoder as,

$$D(W_{1,1}^l; \Theta) = \text{IWT}\left( \left[ \begin{array}{c|c} W_{1,1}^l & \mathbf{0} \\ \hline \mathbf{0} & \mathbf{0} \end{array} \right] \right) + f_\theta(W_{1,1}^l). \tag{2}$$

This design not only allows us to directly input wavelet-encoded images in the decoder, but also lets us use the carefully balanced GAN-based training of ESRGAN since our decoder still operates in the pixel space to recover the missing higher frequencies. Similarly to ESRGAN, we pre-train SRResNet with only L1 Loss, initialize the generator at this state, and then start the full adversarial training with perceptual loss and adversarial loss. We refer to these two models as **ResNet-W** and **ESRGAN-W**, respectively. We defer further training details of the decoder to the Appendix C.2.

**UNet-Decoder** While the ESRGAN-W decoder works well with wavelet-encoded inputs, it does not take full advantage of the compression that wavelet space modeling brings about. Therefore, we also introduce a UNet-based decoder that takes full advantage of the wavelet decomposition of 2D data to completely bypass the original dimensionality of the target image. Due to space limitations, we defer the details of this design to the Appendix C.3.

### 3.1.3 SAMPLER: LEARNED PRIOR NETWORK

The functional prior imposed by our deterministic encoder leads to a highly structured representation space made up of low frequency TL patches of images. In order to generate from NSB-GAN-W, one must be able to draw samples from this space. We posit that, compared to sampling from equivalently-sized representation spaces for AE and VAEs, it is easier to sample from a low-dimensional, image-like latent space using generative methods such as GANs, as they repeatedly have been shown to excel in learning to samples from image distributions. Therefore, we train a BigGAN sampler on this representation space. As the dimensionality is considerably lower than the original image, it only takes a single week and two Tesla V100 GPUs to to train the sampler. Since the values of the TL patch may not lie in the $[0, 1]$, range, care needs to be taken in order to normalize the encoded samplers properly as failing to do so prevents the BigGAN sampler to learn the correct distribution.

**Pre-trained Sampler.** It is possible to use a pre-trained BigGAN model at a lower resolution within NSB-GAN-W to generate higher resolution images. Consider a pre-trained BigGAN model that can generate $128 \times 128$ samples in the pixel-space. By simply projecting the samples into the frequency-space using our deterministic encoder (down to $64 \times 64$) and then up-scaling it through our decoder network, one can generate samples at $256 \times 256$ without any significant loss in quality. In fact, these samplers can outperform the end-to-end baseline BigGAN model not only in compute requirement, but also in terms of the FID at $512 \times 512$ resolution.

### 3.2 NSB-GAN-P

NSB-GAN-P is the pixel counterpart of NSB-GAN-W. Instead of encoding, generating, and decoding in the wavelet space, it operates in the pixel space. The deterministic encoder sub-samples the original image in the pixel space to generate a low-resolution latent representation. Specifically, we realize the sub-sampling as smoothing with a Gaussian filter and sub-sampling with respect to the upscaling factor $r$ (Jo et al., 2018). For the NSB-GAN-P decoder, we use the ESRGAN model with the generator replaced with SRResNet without batch normalization, as in NSB-GAN-W, which we refer to as **ESRGAN-P** and the BigGAN model on low-dimensional space for the prior network. The training procedure is same as in ESRGAN-W, where first we pre-train SRResNet with only L1 loss (namely, **ResNet-P**) and then start adversarial training. Similar to the NSB-GAN-W model, we also have a UNet-version of the decoder but it reconstructs directly in the original data dimensionality like the ESRGAN based decoder. Further details are in Appendix C.2.

### 3.3 TRAINING

As before, let $X$ be the dataset of high-dimensional natural images. The NSB-GAN encoder can be defined as a deterministic function $\mathcal{E} : \mathbb{R}^D \mapsto \mathbb{R}^K$ that uses the WT to produce $Z = \{z_j | \forall j \in \{1, \ldots N\}, z_j \in \mathbb{R}^K\}$. Using this paired dataset $\{X, Z\}$, we treat NSB-GAN as a fully observable model and train the decoder function $D_\theta : \mathbb{R}^K \mapsto \mathbb{R}^D$ to reconstruct $X$ from $Z$ by minimizing the negative log-likelihood (NLL) $-\mathbb{E}_{p(x,z)}[\log p_\theta(x|z)]$ of the conditional probability distribution that it parameterizes. Learning of the generator function $G_\phi : \mathbb{R}^K \mapsto \mathbb{R}^K$ which is referred to as *learning the prior* in previous literature (Razavi et al., 2019; De Fauw et al., 2019), is essentially fitting a generative model to the marginal distribution of $Z$, i.e. $p(z)$. While we could use a similar approach as above and fit $G_\phi$ that parameterizes the model distribution $p_\phi(z)$ by minimizing $-\mathbb{E}_{p(z)}[\log p_\phi(z)]$, we chose to instead use a BigGAN model to directly minimize the (variational lower-bound to the) $f$-divergence $\mathbb{D}_f[p(z)\|p_\phi(z)]$ (Nowozin et al., 2016) between the true marginal distribution $p(z)$ and the model distribution as it provides a better fit for natural-image like distributions compared to other approaches. All together, NSB-GAN simply specifies a generative model over $X$ and $Z$ jointly and is, therefore, trained by minimising the NLL,

$$- \mathbb{E}_{p(x,z)}[\log p_{\theta,\phi}(x, z)] = -\mathbb{E}_{p(x,z)}[\log p_\theta(x|z)] - \mathbb{E}_{p(z)}[\log p_\phi(z)]. \qquad (3)$$

| Sampler | Decoder | Resolution | min FID/IS | Compute Budget |
|---|---|---|---|---|
| Learned-P-64 | ESRGAN-P | 256 | 32.66 / 89.81 | 8 days on $4 \times$ V100 GPUs |
| **Learned-W-64** | **ESRGAN-W** | **256** | **21.82 / 119.8** | **7 days on $4 \times$ V100 GPUs** |
| Learned Baseline | None | 256 | $> 200$ / $< 10$ | 7 days on $8 \times$ V100 GPUs |

Table 1: The learned samplers above are all realized with BigGAN/BigGAN-deep architecture. On a small compute budget, it is not only possible to train the Learned-W-64 sampler well, but also the NSB-GAN-W model (with Learned-W-64 sampler) beats the state-of-the-art VQVAE-2 model (before truncation/rejection sampling), which has an extremely high cost of training similar to that of the BigGAN model. We defer a more detailed discussion about truncation and rejection sampling in Appendix B.1 and showcase a more detailed set of results in Table 4 in the Appendix B.1.

## 4 EXPERIMENTS

In this section, we quantitatively and qualitatively evaluate the training efficiency of the NSB-GAN approach against the end-to-end training approach of the BigGAN model by benchmarking the compute budget required vs. the quality of the generated samples. While quantifying image quality remains a challenging problem (Borji, 2019), we report the Frechet Inception Distance (FID) (Heusel et al., 2017) and Inception Score (IS) (Salimans et al., 2016) as proxy measures to allow for direct comparison with BigGAN. All results are reported on the ImageNet dataset (Deng et al., 2009) at two different resolutions, 256×256 and 512×512.

**Compute Budget.** For our main experiments on training efficiency, depending on the sampler used, we define two different compute budgets. For the learned samplers, we train the NSB-GAN-W and NSB-GAN-P models for a total of 168 hours (7 full days). Our training is performed on a single machine with four Telsa V100 GPUs with 16GB of VRAM each. We found that training the baseline model (BigGAN) at 256x with only 4 GPUs in 7 days is not possible on the default setting, so we allow a total of 8 GPUs for the baseline training. Since the training of these BigGAN-based samplers are highly unstable (Zhao et al., 2020), we train 5 instances in total (3 BigGAN-deep and 2 BigGAN samplers) and report results using the instance with the best FID score. For the pre-trained sampelrs, the compute budget is computed in terms of the TPUs used in their original training and does not include the decoder training. This is because the compute budget required to train the decoder is negligible compared to that of the sampler.

**Hyperparameters and Setup.** For the deterministic encoder using wavelet transformation, we instantiate it with biorthogonal 2.2 wavelet. For the NSB-GAN sampler, we use a batch size of 512 and learning rates of $10^{-4}$ and $4 \times 10^4$ for the generator and discriminator, respectively. For the pre-training of ESRGAN-W and ESRGAN-P with L1 loss, we use a batch size of 32, and the learning rate is initialized $1 \times 10^{-4}$ and decayed by a factor of 2 every $2 \times 10^5$ mini-batch updates. During adversarial training of ESRGAN-W and ESRGAN-P, the learning rate is set to $1 \times 10^{-4}$ and halved at $[50k, 100k]$ iterations. In the case of the UNet-decoders, for the first-level decoder, we use a batch size of 128 and a learning rate of $10^{-4}$. For the second-level decoder, we use the same learning rate as the first-level decoder, but a smaller batch size of 64. We allocate two GPUs to the NSB-GAN sampler, and the other two for the decoders. For the UNet-decoder, this implies one GPU per level. Each component is trained in parallel and independently from each other. To train the BigGAN baseline model at the native resolution of $256 \times 256$, we use the same learning rates as for the NSB-GAN sampler and a recommended batch size of 2048 (Brock et al., 2018). The baseline model trains with eight GPUs, instead of four like with NSB-GAN, to meet its large memory requirement, given the batch size and resolution of the image. We provide training and evaluation code for NSB-GAN sampler and the other models, respectively here: `https://anonymous.4open.science/r/ca7a3f2e-5c27-48bd-a3bc-2dceadc138c1/`.

### 4.1 RESULTS

#### 4.1.1 LEARNED SAMPLERS

To establish the training efficiency and competitive image quality of our NSB-GAN approach, we compare the FID and IS that the NSB-GAN-W, NSB-GAN-P and baseline BigGAN models reach in the compute budget of 7 days with four (8 for the baseline) Tesla V100 GPUs in Table 1. Both NSB-GAN-P and NSB-GAN-W models reach very competitive FIDs of 32.66 and 21.82, respectively. To put these results in perspective, even SAGAN (Zhang et al., 2018a), which requires twice the compute and operates on half the resolution ($128 \times 128$), reaches an FID of $\sim 19$. Furthermore, in the same amount of time but twice the compute, the baseline BigGAN model fails to generate any meaningful samples across all five runs. This illustrates that, compared to end-to-end models, the NSB-GAN models are significantly more compute efficient and can reach competitive image quality.

In the aforementioned models, we use the public PyTorch implementation of BigGAN (kindly provided by the authors) to provide fair comparisons between the NSB-GAN models and the baselines. It is important to note, however, that the Pytorch implementation is not optimal as it does not use Sync-BatchNorm (SBN) which is essential for large mini-batch training. This implementation has been reported to not reach the FID and IS reported in the original paper (ajbrock, 2018).

When the sampler is trained in the low-dimensional space of $64 \times 64$, NSB-GAN-W clearly outperforms NSB-GAN-P. We posit that this sharp difference in FID stems from the difference in the loss of structural information using the two down-sampling methods: pixel-based interpolation and wavelet-space encoding (Sekar et al., 2014). As illustrated in Figure 2 in Appendix A, WT encoding seems to preserve more structural information than pixel-based interpolation which misses key features of the image due to arbitrary sub-sampling. This finding suggests that the down-sampled distribution at $64 \times 64$ becomes sufficiently different from the original distribution at $256 \times 256$ such that when trained on the down-sampled distribution, NSB-GAN-P sampler fails to sufficiently approximate the structure in the original data distribution. If this were true, the NSB-GAN-P performance should improve as the level of down-sampling is decreased. With the following set of experiments with pre-trained samplers, we test this hypothesis and demonstrate that NSB-GAN framework can also be used with pre-trained samplers to make image generation at high resolution ($256 \times 256, 512 \times 512$) efficient and even beat the BigGAN baseline model in some cases.

#### 4.1.2 PRE-TRAINED SAMPLERS

We consider two pairs of pre-trained BigGAN samplers combined with our NSB-GAN models to ultimately generate at two different resolutions of $256 \times 256$ and $512 \times 512$. To clarify, our pre-trained samplers are simply a serial combination of pre-trained BigGAN models [1] and our encoders. For example, Pretrained-128-64 is created by generating samples from a pre-trained sampler at $128 \times 128$ and then applying a down-sampling operation (pixel or wavelet) to obtain $64 \times 64$ samples. Then, our decoders up-scale the low-resolution image four times to $256 \times 256$. We report the FID/IS scores for all the samplers and compare to the baseline BigGAN models pre-trained at the respective resolutions in Table 2. First, notice that NSB-GAN-P performance increases drastically, supporting our hypothesis. Next, both NSB-GAN-W and NSB-GAN-P models require significantly less compute (up to 4 times less TPU-v3 cores, as shown in the last row of Table 2) to train. Most importantly, note how approach outperforms the original BigGAN model in terms of FID at $512 \times 512$ resolution, despite using exactly half the original compute budget. We hypothesize that this is partly due to the way in which ImageNet $512 \times 512$ dataset is generated, but defer this discussion to Appendix H.

Furthermore, note that we did not have to re-train the decoders when applying to different resolutions and samplers. All results in this experiment are done with the decoders from the previous experiment that were only trained to up-scale images from $64 \times 64$ to $256 \times 256$. They evidently generalize well across different resolutions. This amortization of training across resolutions further reduces the training cost when more than one generator is trained.

**UNet-Decoders** As shown in Table 4 of the Appendix B.1, our ESRGAN-based decoders clearly outperform our UNet-decoders on FID and IS. It is important to note, however, that this difference is primarily because the ESRGAN decoders employ adversarial training which has been shown to drastically improve image quality as measured by FID. As shown in Table 3, not considering

---

[1]We use the pre-trained models from huggingface's PyTorch re-implementation of BigGAN model.

| Sampler | Decoder | Resolution | min FID / IS | Compute |
|---|---|---|---|---|
| BigGAN-128 | None | 128 | 10.58 / 43.72 | 128 TPUs |
| BigGAN-256 | None | 256 | 10.68 / 52.16 | 256 TPUs |
| Pretrained-128-64 | ESRGAN-P | 256 | 12.28 / 46.06 | 128 TPUs |
| Pretrained-128-64 | ESRGAN-W | 256 | 12.66 / 45.54 | 128 TPUs |
| BigGAN-512 | None | 512 | 11.32 / 49.37 | 512 TPUs |
| **Pretrained-256-128** | **ESRGAN-P** | **512** | **10.30 / 213.4** | **256 TPUs** |
| **Pretrained-256-128** | **ESRGAN-W** | **512** | **10.59 / 52.14** | **256 TPUs** |
| Pretrained-128 | ESRGAN-P | 512 | 13.55 / 47.70 | 128 TPUs |

Table 2: All pretrained samplers above are realized with BigGAN. In higher dimensions, compared to the baseline BigGAN models at the respective resolutions, NSB-GAN models reduce the training compute budget by up to four times (last row) while suffering only a minor increase in FID. All pre-trained samplers above are trained for approximately two days on the compute described. Note that the decoder is only trained once, but generalizes across all the resolutions. This amortization further reduces the training cost drastically. Pretrained-128-64, for example, indicates that the model generates at $128 \times 128$ resolution and we down-sample it to $64 \times 64$ resolution with our encoders for NSB-GAN models. As empirically tested and confirmed by Razavi et al. (2019), IS is highly sensitive to slight blurs and perturbations. Therefore, we include an expanded set of quantitative results with various truncation and rejection sampling levels in Table 5 in the Appendix B.2.

|  | UNet-P | UNet-W | VQVAE-2 |
|---|---|---|---|
| **Train MSE** | 0.0061 | **0.0045** | 0.0047 |
| **Valid MSE** | 0.0074 | **0.0049** | 0.0050 |

Table 3: MSE on training and validation set for UNet-P, UNet-W,and VQVAE-2 models. Small difference between the training and validation error suggests that the models generalize well.

adversarial decoder, our UNet-W decoder leads to the best reconstruction error compared to the pixel-space decoder, including the state-of-the-art VQVAE-2 model.

**Evaluation on LSUN Church with StyleGAN-2-W**  The NSB-GAN approach can be implemented with models other than BigGAN. To demonstrate this, we replace the BigGAN sampler architecture in our model with the StyleGAN-2 architecture and re-run the experiments on the LSUN-Church dataset. The results in Table 10 of the Appendix E demonstrate that our NSB-GAN models can generalize to different samplers and datasets.

**Qualitative Results.**  Due to space limitations, we have deferred all qualitative results to the Appendix. Please see the Appendix A.1, A.2, A.3, and A.4 for a detailed qualitative comparison of all the models.

## 5 CONCLUSION

In this work, we present a new genre of compute-efficient generative models, NOT-SO-BIG-GAN, that achieve comparable image quality to the current SoTA DGM (BigGAN) with a dramatically lower compute budget. Surprisingly, NOT-SO-BIG-GAN is even able to outperform BigGAN in image quality at $512 \times 512$ resolution. Overall, we hope that our work inspires others to develop low-compute generative models that can be utilized and iterated on by the wider research community.

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

## A    APPENDIX

### A.1    NSB-GAN SAMPLES WITH LEARNED SAMPLER

Refer to Figure 3 for samples from NSB-GAN models.

### A.2    NSB-GAN SAMPLES WITH PRE-TRAINED SAMPLER

Refer to Figures 4b and 4a for class-conditional samples from NSB-GAN-P and NSB-GAN-W models at $256 \times 256$, respectively. Refer to Figures 7a, 8a, 5a, and 6a for class-conditional samples from NSB-GAN-P and NSB-GAN-W models at $512 \times 512$. Samples from BigGAN models at $256 \times 256$ and $512 \times 512$ are also included in 9, 10, and 11 for comparison. We intentionally include random samples from the same classes for all the models to allow for direct and easy comparison.

### A.3    FULL RESOLUTION SAMPLES FROM NSB-GAN-W WITH PRE-TRAINED SAMPLER

Refer to Figures 12, 13, 14, and 15 for full resolution samples from NSB-GAN-W at $256 \times 256$.

Refer to Figures 16, 17, 18, and 19 for $512 \times 512$. These samples are fitted to the page.

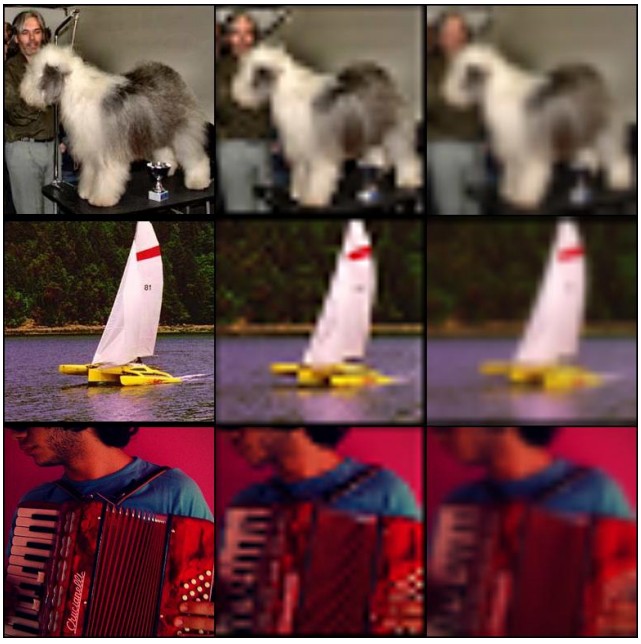

Figure 2: The above images are the original, down-sampled version using wavelet encoding, and down-sampled version using pixel-based interpolation (bilinear) in the column-wise order. Please note that the pixel-based interpolation loses key structural information in the images – face of the person, structure of the boat, and accordion keyboard – whereas wavelet encoding does not.

| Sampler | Decoder | Resolution | min FID / IS | FID / max IS |
|---|---|---|---|---|
| Learned-P-64 | ESRGAN-P | 256 | 32.66 / 89.81 | 33.61 / 96.12 |
| Learned-W-64 | ESRGAN-W | 256 | 21.82 / 119.75 | 27.52 / 219.4 |
| Learned-P-64 | UNet-W | 256 | 35.60 / 84.68 | 38.58 / 129.4 |

Table 4: Minimum FID / IS (column 4) and FID / Minimum IS (column 5) attained with different levels of truncation [1.0, 0.8, 0.6, 0.4, 0.2, 0.1] and rejection sampling [0.70, 0.80, 0.90, 0.95].

### A.4 FULL RESOLUTION SAMPLES FROM NSB-GAN-P WITH PRE-TRAINED SAMPLER

Refer to Figures 20, 21, 22, and 23 for full resolution samples from NSB-GAN-W at $256 \times 256$.

Refer to Figures 24, 25, 26, and 27 for $512 \times 512$. These samples are fitted to the page.

## B RESULTS WITH TRUNCATION AND REJECTION SAMPLING

### B.1 LEARNED SAMPLERS

For a more thorough analysis of our NSB-GAN models with learned samplers, we apply truncation and rejection sampling to study their effects on FID and IS. We apply truncation at [1.0, 0.8, 0.6, 0.4, 0.2, 0.1], and, given the truncation level at which the model outputs the best FID, we apply rejection sampling with a pre-trained Inception V3 model on ImageNet. We test rejection sampling thresholds at 0.70, 0.80, 0.90, and 0.95. Refer to Table 4 for quantitative results on FID and IS. We also include FID and IS results for UNet-W decoder that was only trained with a MSE loss.

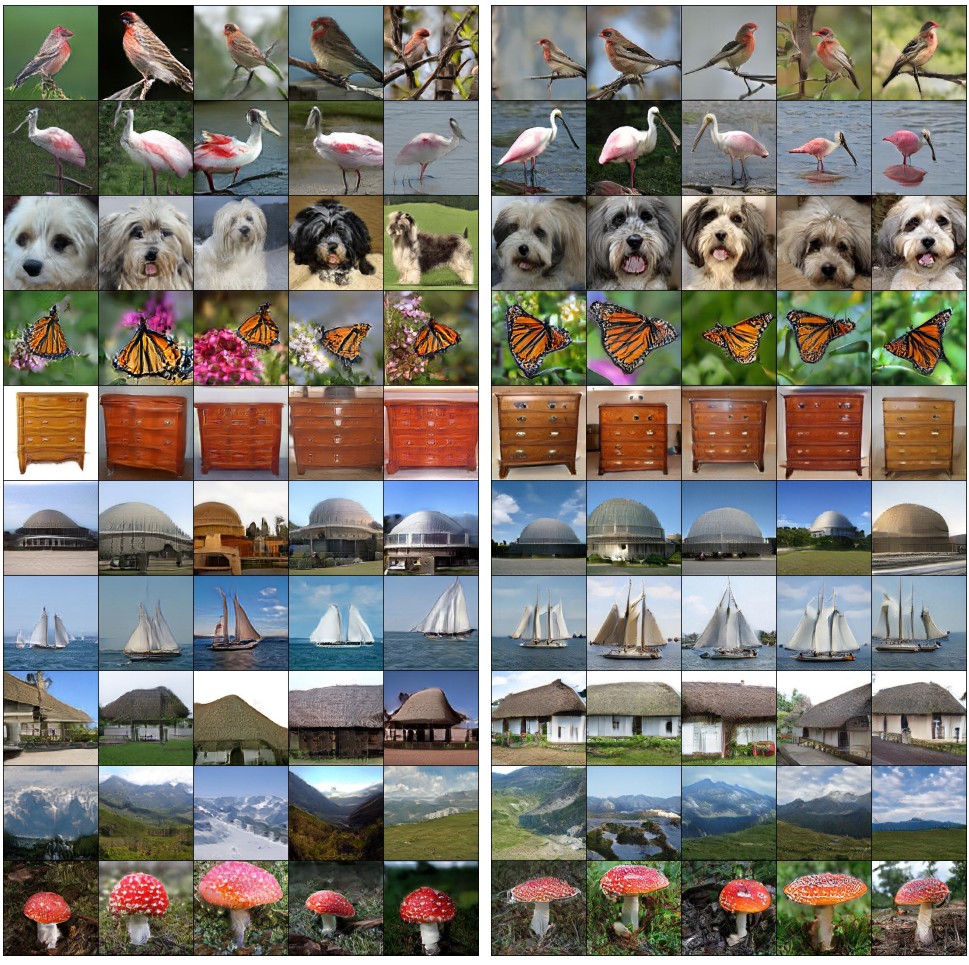

(a) NSB-GAN-P with learned sampler          (b) NSB-GAN-W with learned sampler

Figure 3: Class-conditional random samples at $256 \times 256$ from NSB-GAN models with learned samplers. Classes from the top row: 12 house finch, 129 spoonbill, 200 Tibetan terrier, 323 monarch butterfly, 493 chiffonier, 727 planetarium, 780 schooner, 853 thatch, 970 alp, and 992 agaric.

| Sampler | Decoder | Resolution | min FID / IS | FID / max IS |
|---|---|---|---|---|
| Pretrained-128-64 | ESRGAN-P | 256 | 12.28 / 46.06 | 13.85 / 229.6 |
| Pretrained-128-64 | ESRGAN-W | 256 | 12.66 / 45.54 | 20.44 / 285.5 |
| Pretrained-256-128 | ESRGAN-P | 512 | 10.30 / 213.35 | 21.85 / 338.4 |
| Pretrained-256-128 | ESRGAN-W | 512 | 10.59 / 52.14 | 22.26 / 332.7 |

Table 5: Minimum FID / IS (column 4) and FID / Minimum IS (column 5) attained with different levels of truncation [1.0, 0.8, 0.6, 0.4, 0.2, 0.1] and rejection sampling [0.70, 0.80, 0.90, 0.95].

## B.2 PRE-TRAINED SAMPLERS

We conduct the same detailed analysis for our NSB-GAN models with pre-trained samplers as the above. Refer to Table 5 for quantitative results on FID and IS.

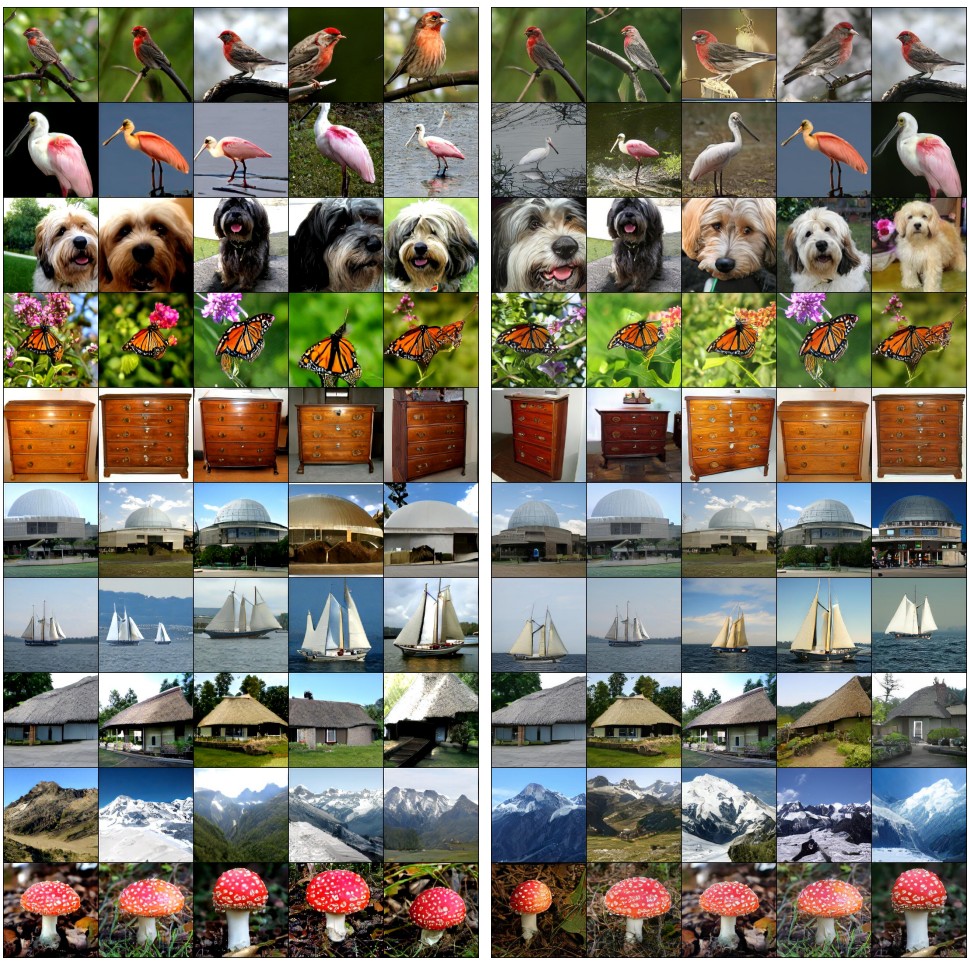

(a) NSB-GAN-W with pre-trained sampler      (b) NSB-GAN-P with pre-trained sampler

Figure 4: Class-conditional random samples at $256 \times 256$ from NSB-GAN models with pre-trained samplers. Classes from the top row: 12 house finch, 129 spoonbill, 200 Tibetan terrier, 323 monarch butterfly, 493 chiffonier, 727 planetarium, 780 schooner, 853 thatch, 970 alp, and 992 agaric.

## C   ARCHITECTURE, HYPERPARAMETERS, AND TRAINING DETAILS

### C.1   SAMPLER

Refer to 6 for details about the architecture and hyperparameters of the learned samplers (Learned-P-64 and Learned-W-64). We attempted to train the sampler with both BigGAN and BigGAN-deep architectures and report on the models that achieved the best FID for each. Empirically, we found the models trained on the pixel-space much more unstable than on the wavelet-space, with four out of five models diverging.

### C.2   NSB-GAN DECODERS

**ESRGAN-W and ESRGAN-P**   A similar training procedure in ESRGAN is conducted for both of our decoders, ESRGAN-W and ESRGAN-P. First, SRResNet, with batch normalization removed, is trained with $L_1$ loss: $L_1 = \mathbb{E}_{x_i} ||G(x_i) - y||_1$, where $x_i$ is the down-sampled image and $y$ is the target image. After training with this $L_1$ loss for 150k iterations, the GAN is trained with perceptual and adversarial losses added. Therefore, the total loss for the generator becomes:

$$L_{total} = L_{percep} + \lambda L_G + \eta L_1 \tag{4}$$

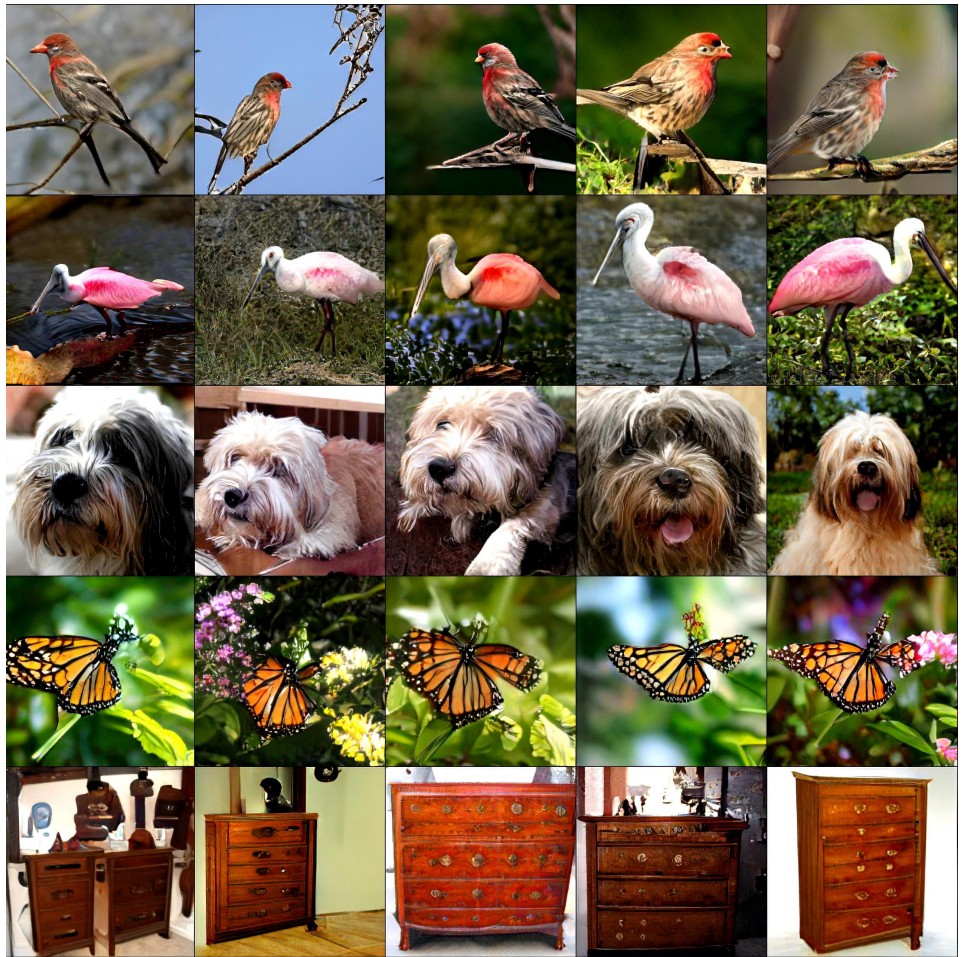

(a) NSB-GAN-W with pre-trained sampler

Figure 5: Class-conditional random samples at $512 \times 512$ from NSB-GAN-W model with pre-trained sampler. Classes from the top row: 12 house finch, 129 spoonbill, 200 Tibetan terrier, 323 monarch butterfly, and 493 chiffonier.

Perceptual loss is implemented with a pre-trained VGG-19 model on ImageNet and we use the features of the fourth layer before the fifth max-pooling layer. This GAN model is then trained for another 150k iterations.

As in the training for the learned sampler in wavelet domain, because the wavelet-encoded input image does not lie in the standard $[0, 1]$ range for ESRGAN-W, a normalization step is applied to transform the range into $[0, 1]$. By using the minimum and maximum values of the wavelet-encoded input images, pre-calculated in the pre-processing step, we use the following normalization technique: $x_{i_{norm}} = \frac{x_i + \lceil |min_x| \rceil}{\lceil max_x \rceil}$, where $x_i$ is the wavelet-encoded input and $min_x$ and $max_x$ are the pre-calculated minimum and maximum values of the wavelet-encoded input images.

Both ESRGAN-W and ESRGAN-P share the same architecture design, except for how the input is up-scaled and added to the learned features in order to induce the residual learning paradigm. Refer to Table 7 for specific details of the architecture and hyperparameters.

### C.3    UNET

**UNet in Wavelet Domain**    After a wavelet transform, the original image can be deterministically recovered from the TL, TR, BL, and BR patches using IWT. NSB-GAN-W, however, discards the high-frequency patches (TR, BL, and BR) during its encoding step, rendering a fully deterministic

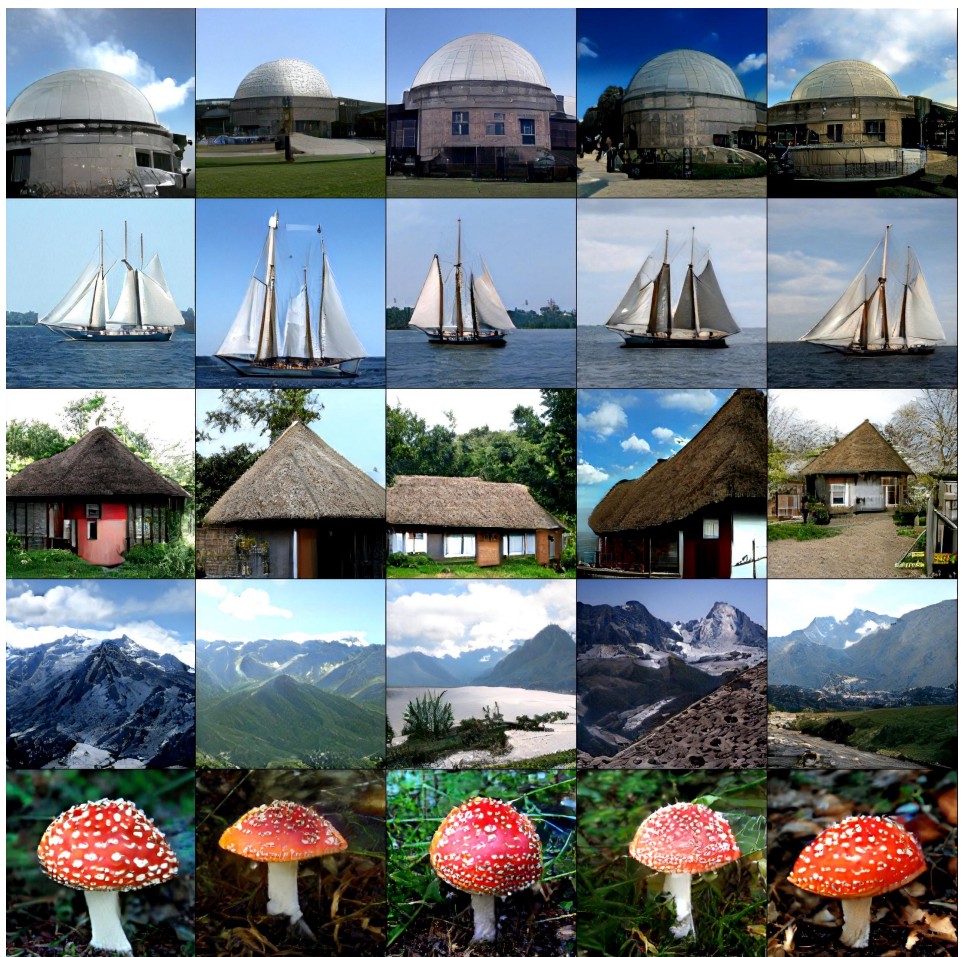

(a) NSB-GAN-W with pre-trained sampler

Figure 6: Additional class-conditional random samples at $512 \times 512$ from NSB-GAN-W model with pre-trained sampler. Classes from the top row: 727 planetarium, 780 schooner, 853 thatch, 970 alp, and 992 agaric.

decoding impossible. To resolve this, the NSB-GAN decoder first learns to recover the missing high-frequency patches (using a neural network) and then deterministically combines them using IWT to reconstruct the original input.

Since NSB-GAN's encoding operation recursively applies wavelet transforms and discards the high-frequency components at each of the $L$ encoding levels, we train $L$ decoder networks to reconstruct the missing frequencies at each corresponding level of decoding. To parallelize the training of these decoder networks, we perform multiple wavelet transforms to the original high-resolution dataset, generating $L$ training sets of TL, TR, BL and BR patches. This allows us to independently train each of the decoder networks (in a supervised manner) to reconstruct the missing high-frequency patches conditioned on the corresponding TL. This parallelization boosts the convergence rate of NSB-GAN, allowing for a fully trained decoder in under 48 hours.

Again with a slight abuse of notation, let $W_{1,1}^l$ be the TL patch at level $l$ and $\mathrm{IWT} = \mathrm{WT}^{-1}$. We can write the decoder for level $l$ as

$$D_l(W_{1,1}^l; \Theta) = \mathrm{IWT}\left( \left[ \begin{array}{c|c} W_{1,1}^l & f_{\theta_{1,2}}^l(W_{1,1}^l) \\ \hline f_{\theta_{2,1}}^l(W_{1,1}^l) & f_{\theta_{2,2}}^l(W_{1,1}^l) \end{array} \right] \right)$$
$$\text{for} \quad 1 \le l \le L. \tag{5}$$

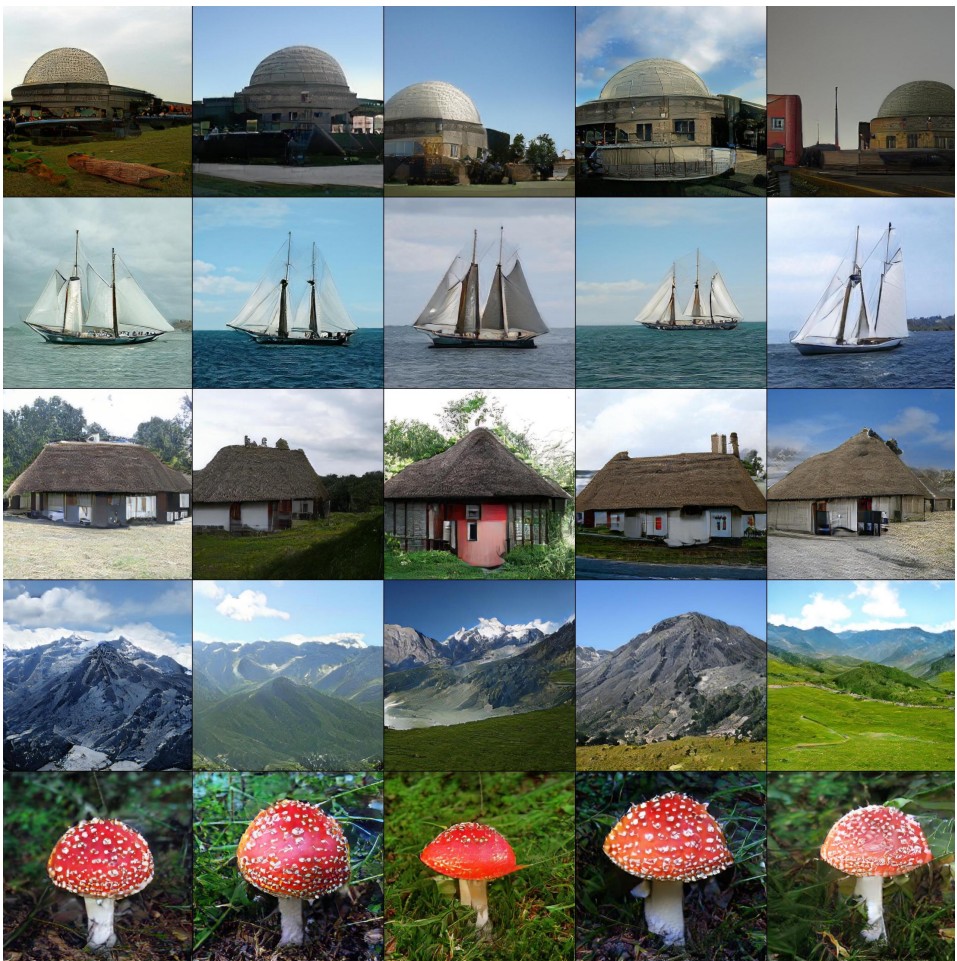

(a) NSB-GAN-W with pre-trained sampler

Figure 7: Class-conditional random samples at $512 \times 512$ from NSB-GAN-P model with pre-trained sampler. Classes from the top row: 12 house finch, 129 spoonbill, 200 Tibetan terrier, 323 monarch butterfly, and 493 chiffonier.

Here, each $f^l$ is a deep neural network that is trained to reconstruct one of the TR, BL, or BR patches at level $l$, conditioned on the TL patch ($W_{1,1}^l$).

The main challenge in high-resolution image generation lies in overcoming the curse of dimensionality. But by leveraging IWT, the original dimensionality of the image is completely bypassed in the NSB-GAN decoder. Therefore, in practice, at each level, we further divide the TR, BL and BR patches by applying WT until we reach the patch dimensionality of 32x32. This can be done irrespective of the original dimensionality of the image. We then reconstruct these patches in parallel and recover the patches and the original image by recursively applying IWT. This ability of NSB-GAN to bypass the original dimensionality of the input separates it from all other SR methods that operate in the pixel space.

Refer to Figure 28a and Table 8 for architecture details and hyperparameters of the UNet decoder for the wavelet-space.

Refer to Figure 30 for the training schematic with the UNet decoders in the wavelet-space.

**Architecture**    We realize each of the $L$ decoder neural networks $f_{\theta_l}$ with a slightly modified version of the UNet architecture (Ronneberger et al., 2015) rather than the commonly used transposed-convolution based architecture. UNet is typically used for image segmentation and object detection tasks. As shown in Figure 28, UNet is an autoencoder architecture that has skip-connections from

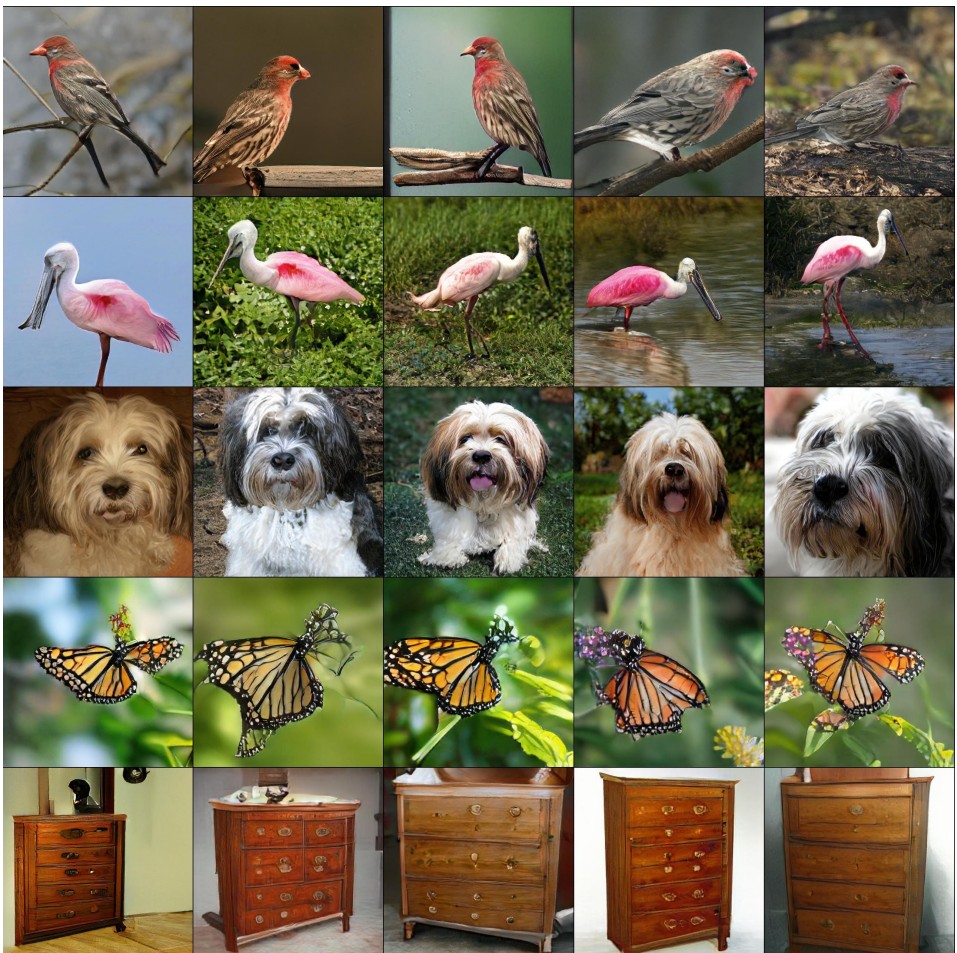

(a) NSB-GAN-W with pre-trained sampler

Figure 8: Additional class-conditional random samples at $512 \times 512$ from NSB-GAN-P model with pre-trained sampler. Classes from the top row: 12 house finch, 129 spoonbill, 200 Tibetan terrier, 323 monarch butterfly, and 493 chiffonier.

each encoding layer to its corresponding decoding layer. These skip connections copy and paste the encoding layer's output into the decoding layer, allowing the decoder to exclusively focus on reconstructing only the missing, residual information. This architectural design makes it a compelling fit for decoding in NSB-GAN. We modify the UNet architecture by appending three shallow networks to its output with each one reconstructing one of the three high-frequency patches. This setup allows us to capture the dependencies between the high-frequency patches while also allowing sufficient capacity to capture to their individual differences.

**UNet in Pixel Domain** The UNet-P decoder is also a partly-learned function that uses a modified UNet-based architecture. First, the encoded image is deterministically upsampled using an interpolation-based method. This leads to a low-quality, blurry image at the same size as the original image. We then train a UNet to fill in the missing details in a similar approach to image super-resolution methods (Hu et al., 2019). Unlike the NSB-GAN decoder that circumvents the CoD by avoiding reconstructions in the original data dimensionality, the NSB-GAN-P decoder still has to operate on the full image size, resulting in a larger decoder.

Refer to Figure 28b and Table 9 for architecture details and hyperparameters of the UNet decoder for the pixel-space.

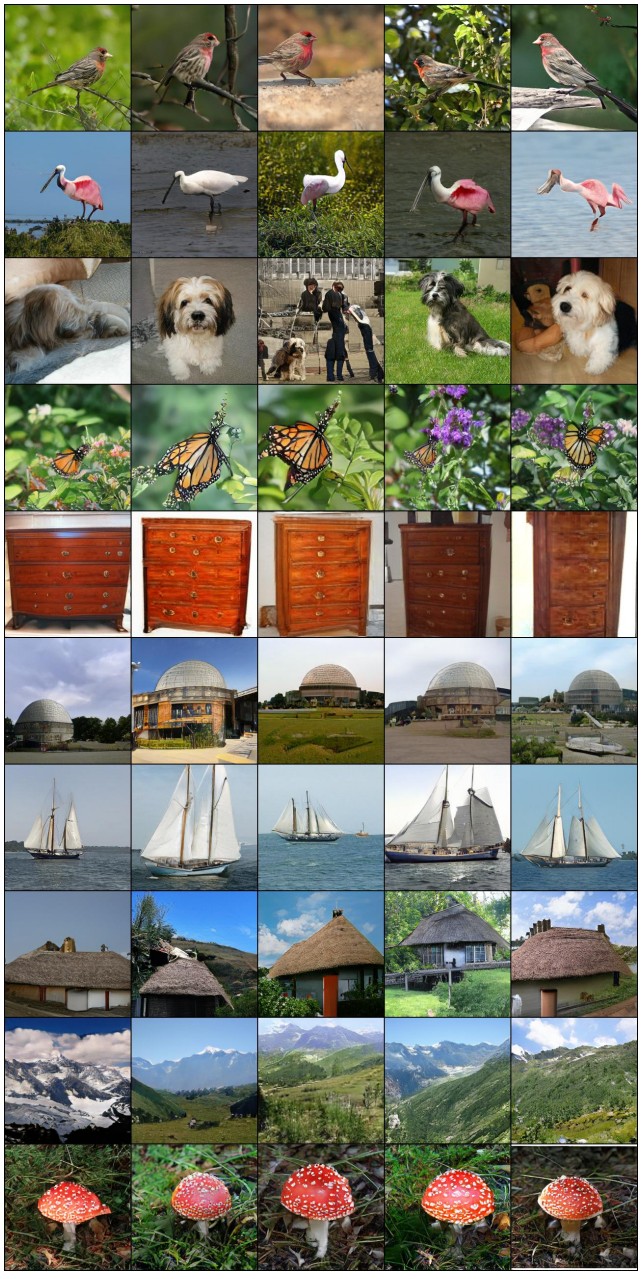

Figure 9: Class-conditional random samples at $256 \times 256$ from BigGAN. Classes from the top row: 12 house finch, 129 spoonbill, 200 Tibetan terrier, 323 monarch butterfly, 493 chiffonier, 727 planetarium, 780 schooner, 853 thatch, 970 alp, and 992 agaric.

# D    NSB-GAN VS NSB-GAN-P INFORMATION CONTENT

As we show in Figure 2, the information content in the latent embedding of NSB-GAN and NSB-GAN-P are drastically different. The TL patches from NSB-GAN preserve more structural information in the image than the down-sampled image from NSB-GAN-P. It is evident in the figure that pixel-based down-sampling method misses key structures and features, such as the face of the person, structure of the boat, and structure of the accordion keyboard, whereas wavelet encoding does not.

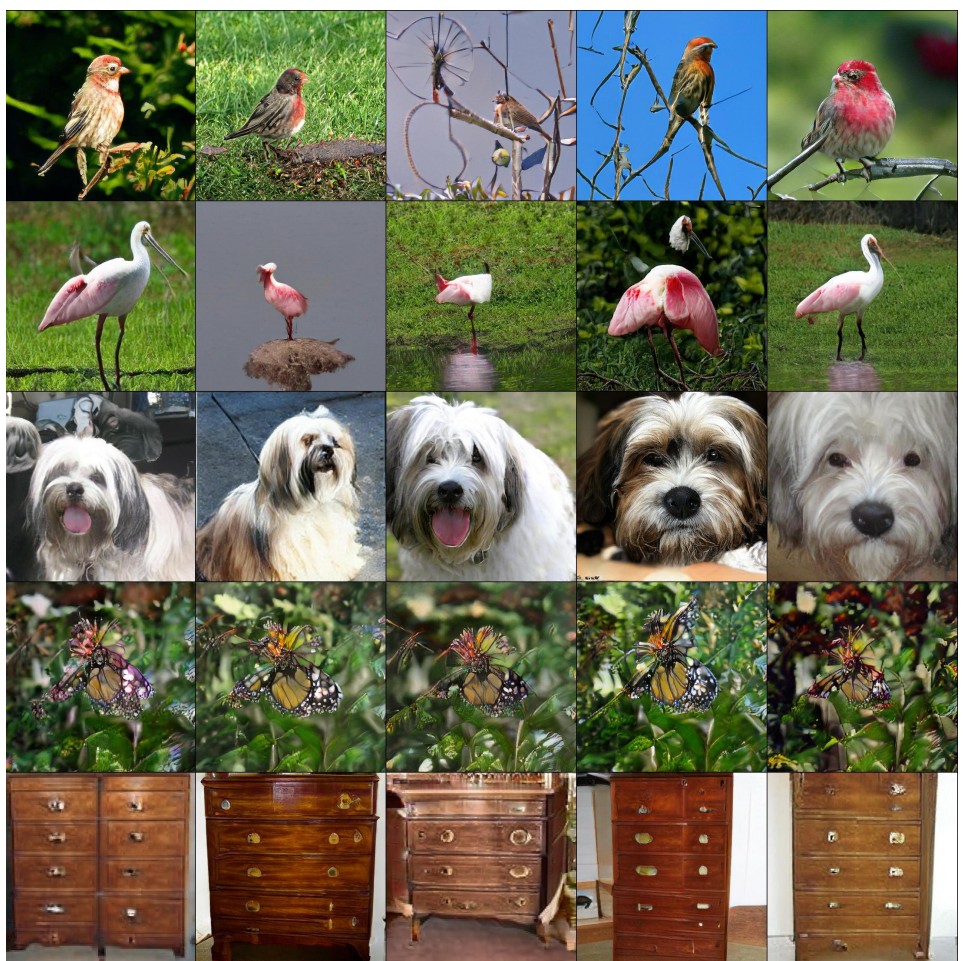

Figure 10: Class-conditional random samples at $512 \times 512$. Classes from the top row: 12 house finch, 129 spoonbill, 200 Tibetan terrier, 323 monarch butterfly, and 493 chiffonier.

|  | Learned-W-64 | Learned-P-64 |
|---|---|---|
| **Best model type** | BigGAN-deep | BigGAN |
| **Batch size** | 512 | 512 |
| **Learning rate of generator** | 1e-4 | 1e-4 |
| **Learning rate of discriminator** | 4e-4 | 4e-4 |
| **Attention resolution** | 128 | 120 |
| **Dimension of random noise ($z$ dim)** | 32 | 32 |
| **Number of resblocks per stage in generator/discriminator** | 2 | 1 |
| **Adam optimizer $\beta_1$** | 0 | 0 |
| **Adam optimizer $\beta_2$** | 0.999 | 0.999 |
| **Adam optimizer $\epsilon$** | 1e-8 | 1e-8 |
| **Training iterations** | 250000 | 250000 |

Table 6: Hyperparameters of learned samplers (wavelet and pixel)

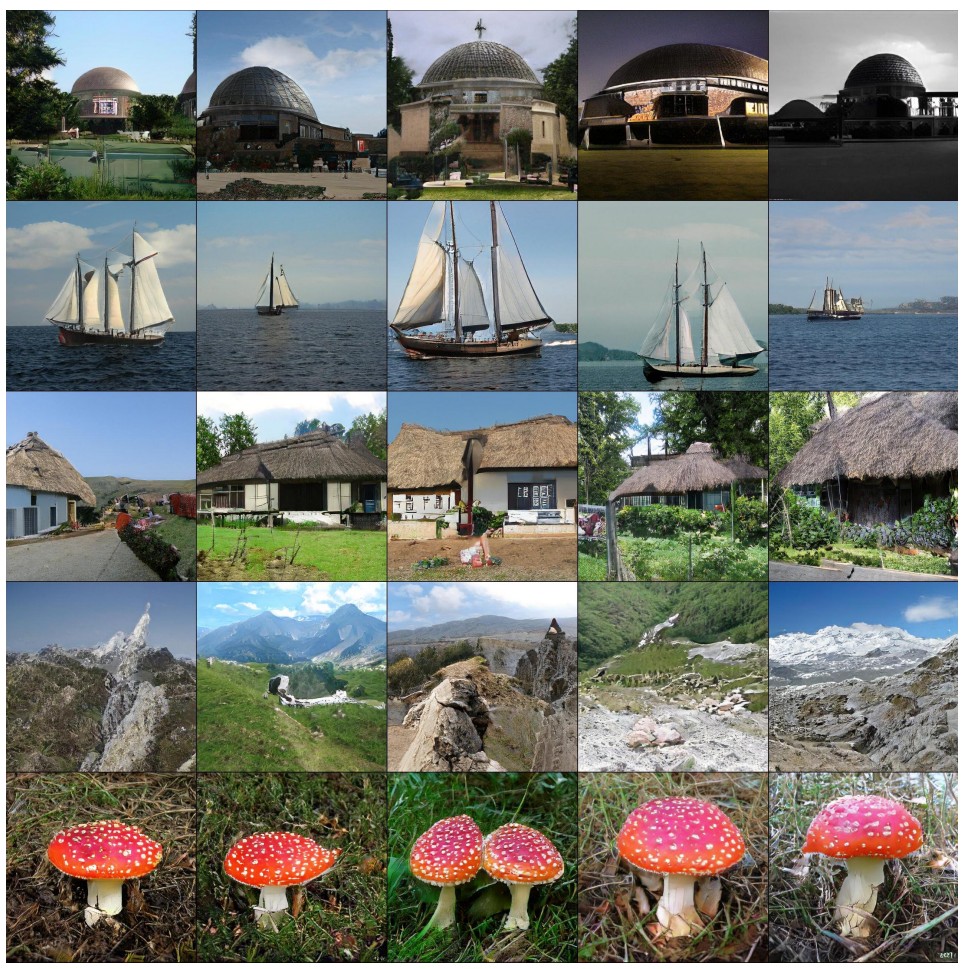

Figure 11: Additional class-conditional random samples at $512 \times 512$. Classes from the top row: 727 planetarium, 780 schooner, 853 thatch, 970 alp, and 992 agaric.

|  | ESRGAN-W / ESRGAN-P |
|---|---|
| **Input size** | $64 \times 64$ |
| **Batch size** | 32 |
| **Learning rate** | 1e-4 |
| **Number of residual blocks** | 16 |
| **Conv filter size** | 3 |
| **Adam optimizer** $\beta_1$ | 0.9 |
| **Adam optimizer** $\beta_2$ | 0.99 |
| $L_1$ **loss weight** ($\eta$) | 1e-2 |
| $L_G$ **loss weight** ($\lambda$) | 5e-3 |
| **Training iterations** | 150000 |

Table 7: Hyperparameters of ESRGAN decoders

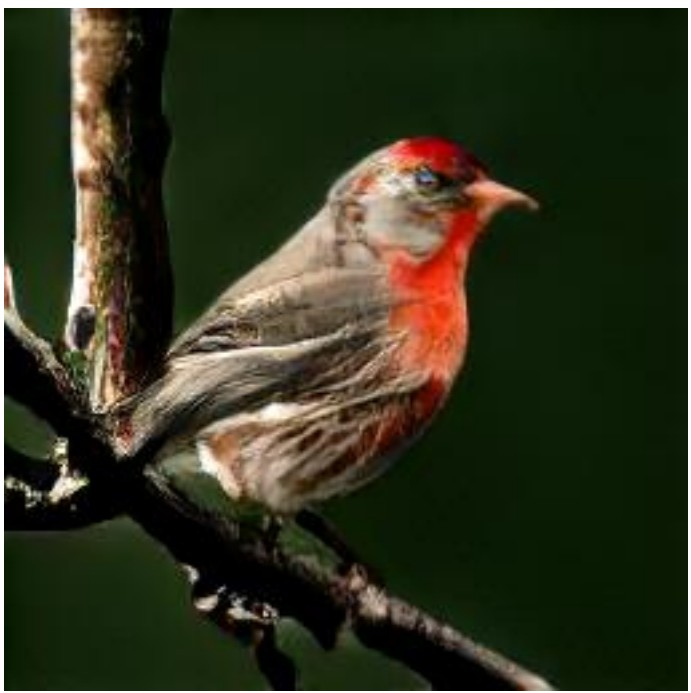

Figure 12: Full $256 \times 256$ resolution sample from NSB-GAN-W with pre-trained sampler.

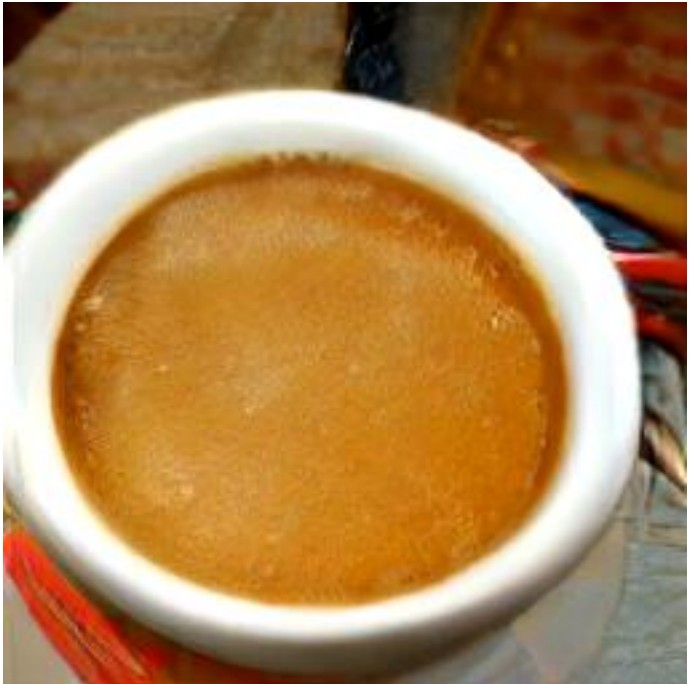

Figure 13: Full $256 \times 256$ resolution sample from NSB-GAN-W with pre-trained sampler.

# E    EVALUATION ON LSUN CHURCH WITH STYLEGAN-2

To demonstrate that the NSB-GAN paradigm can be different samplers and that our decoder trained on ImageNet generalizes fairly well to an unseen dataset, we conduct a set of experiments with the BigGAN sampler replaced with a StyleGAN-2 sampler. We test the StyleGAN-2 sampler with

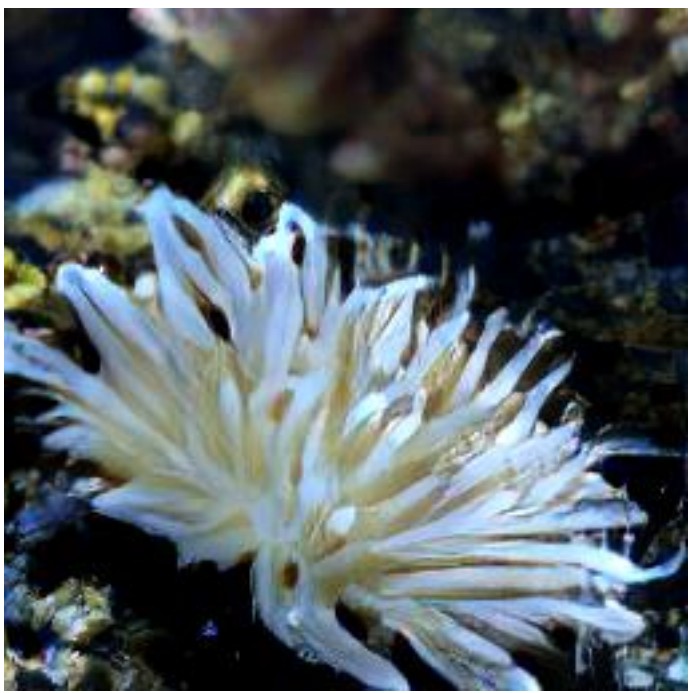

Figure 14: Full $256 \times 256$ resolution sample from NSB-GAN-W with pre-trained sampler.

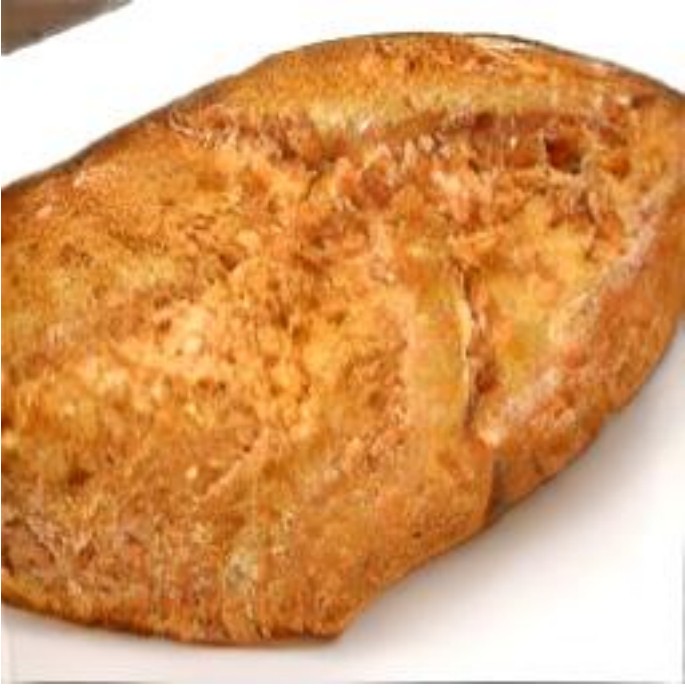

Figure 15: Full $256 \times 256$ resolution sample from NSB-GAN-W with pre-trained sampler.

two decoders: one trained on ImageNet and another trained on LSUN Church. Illustrated in 10, as expected, the decoder that was only trained on the LSUN Church dataset outperforms our decoder trained on ImageNet. However, even with our decoder, we reach a very competitive FID of 13.53, beating other recent two-step approaches, such as (Liu et al., 2019a).

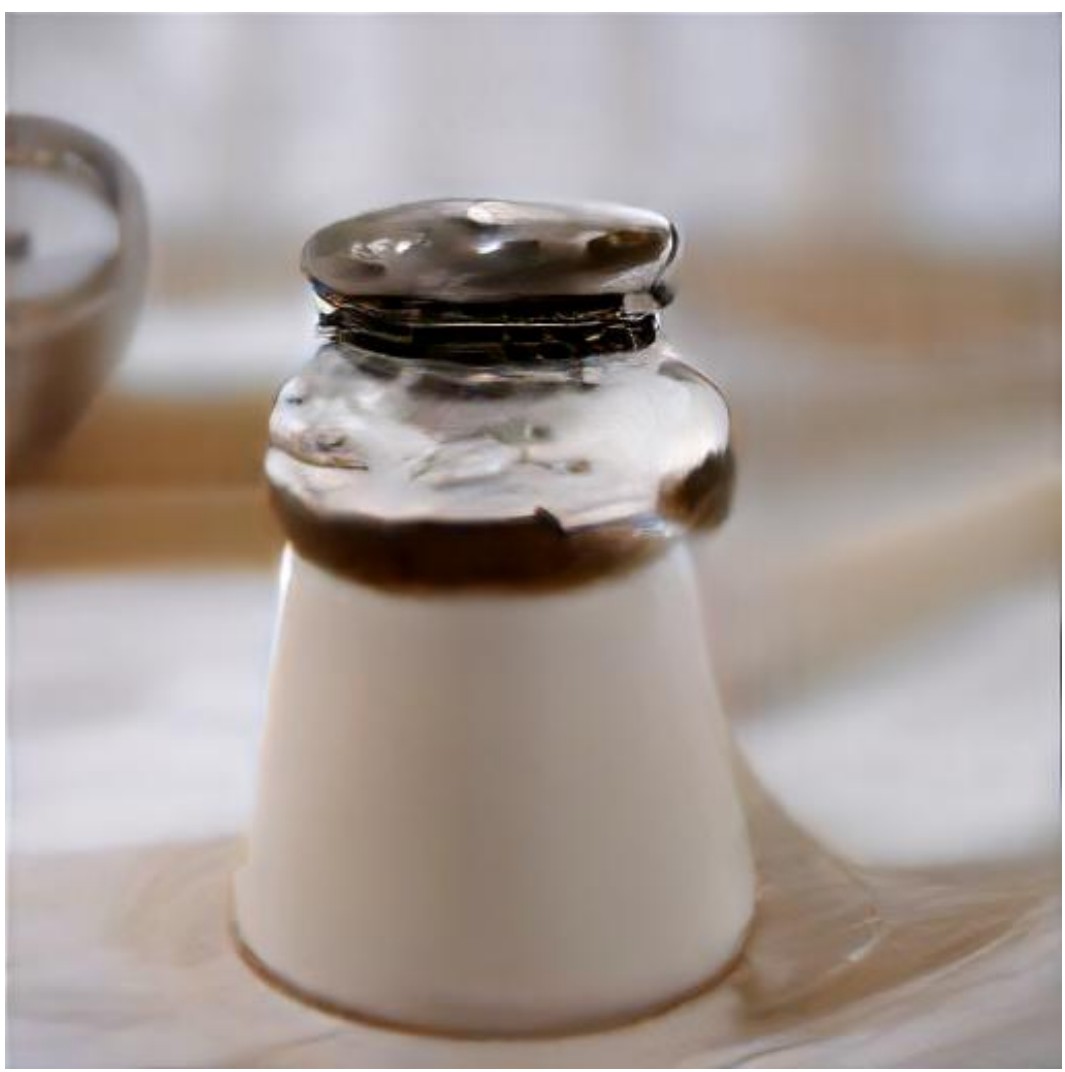

Figure 16: Full $512 \times 512$ resolution sample from NSB-GAN-W with pre-trained sampler.

|                                 | Level 1 Decoder | Level 2 Decoder |
| --- | :---: | :---: |
| **Input size**                  | $32 \times 32$ | $32 \times 32$ |
| **Batch size**                  | 128 | 64 |
| **Learning rate**               | 1e-4 | 1e-4 |
| **Layers**                      | 16 | 16 |
| **Conv filter size**            | 3 | 3 |
| **Number of added shallow networks** | 12 | 48 |
| **Adam optimizer $\beta_1$**    | 0.9 | 0.9 |
| **Adam optimizer $\beta_2$**    | 0.999 | 0.999 |
| **Adam optimizer $\epsilon$**   | 1e-8 | 1e-8 |
| **Training iterations**         | 288000 | 232000 |

Table 8: Hyperparameters of UNet-W decoders (level 1 and level 2)

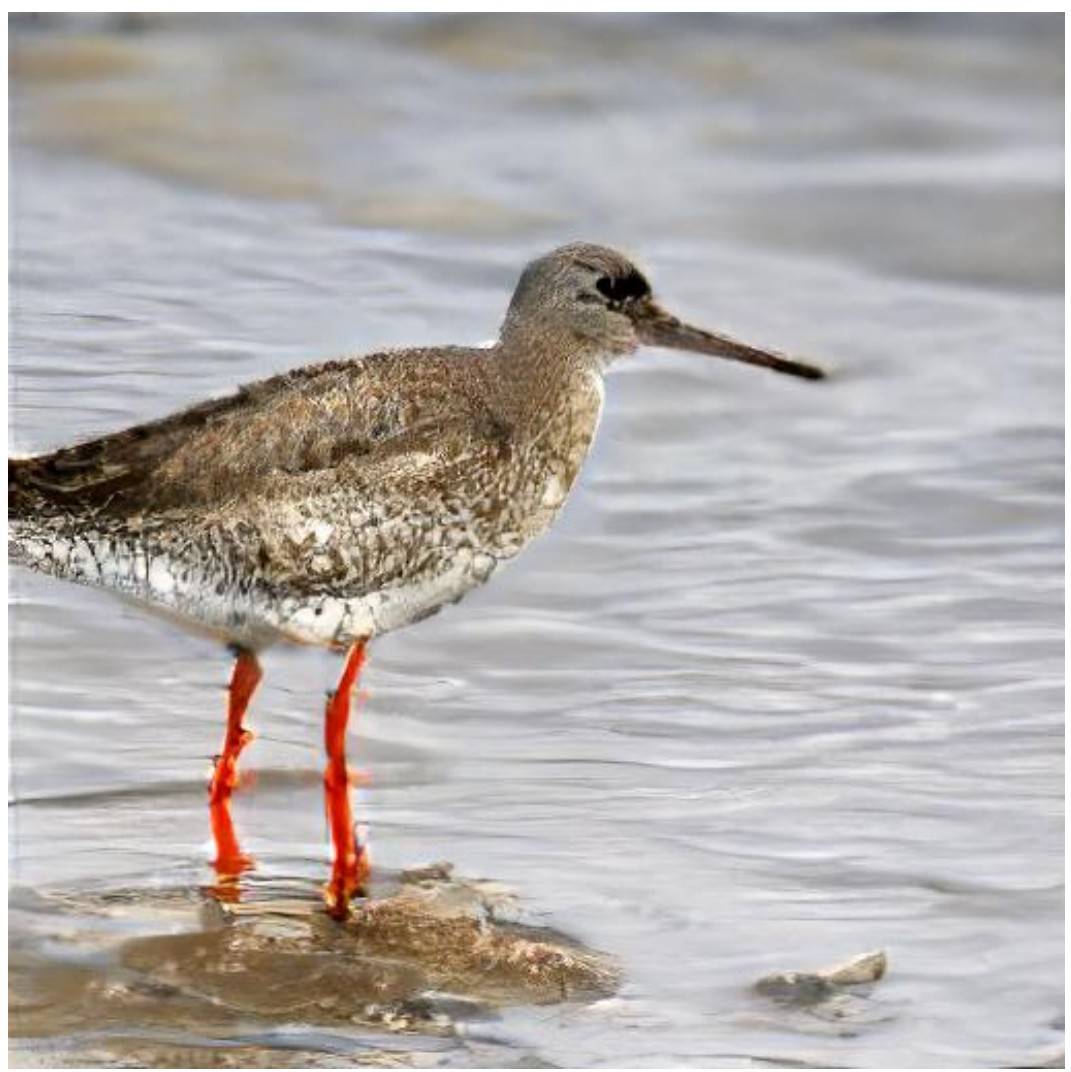

Figure 17: Full $512 \times 512$ resolution sample from NSB-GAN-W with pre-trained sampler.

|  | Level 1 Decoder | Level 2 Decoder |
|---|---|---|
| **Input size** | $64 \times 64$ | $128 \times 128$ |
| **Batch size** | 64 | 64 |
| **Learning rate** | 1e-4 | 1e-4 |
| **Layers** | 19 | 19 |
| **Conv filter size** | 3 | 3 |
| **Adam optimizer $\beta_1$** | 0.9 | 0.9 |
| **Adam optimizer $\beta_2$** | 0.999 | 0.999 |
| **Adam optimizer $\epsilon$** | 1e-8 | 1e-8 |
| **Training iterations** | 615000 | 178000 |

Table 9: Hyperparameters of UNet-P decoders (level 1 and level 2)

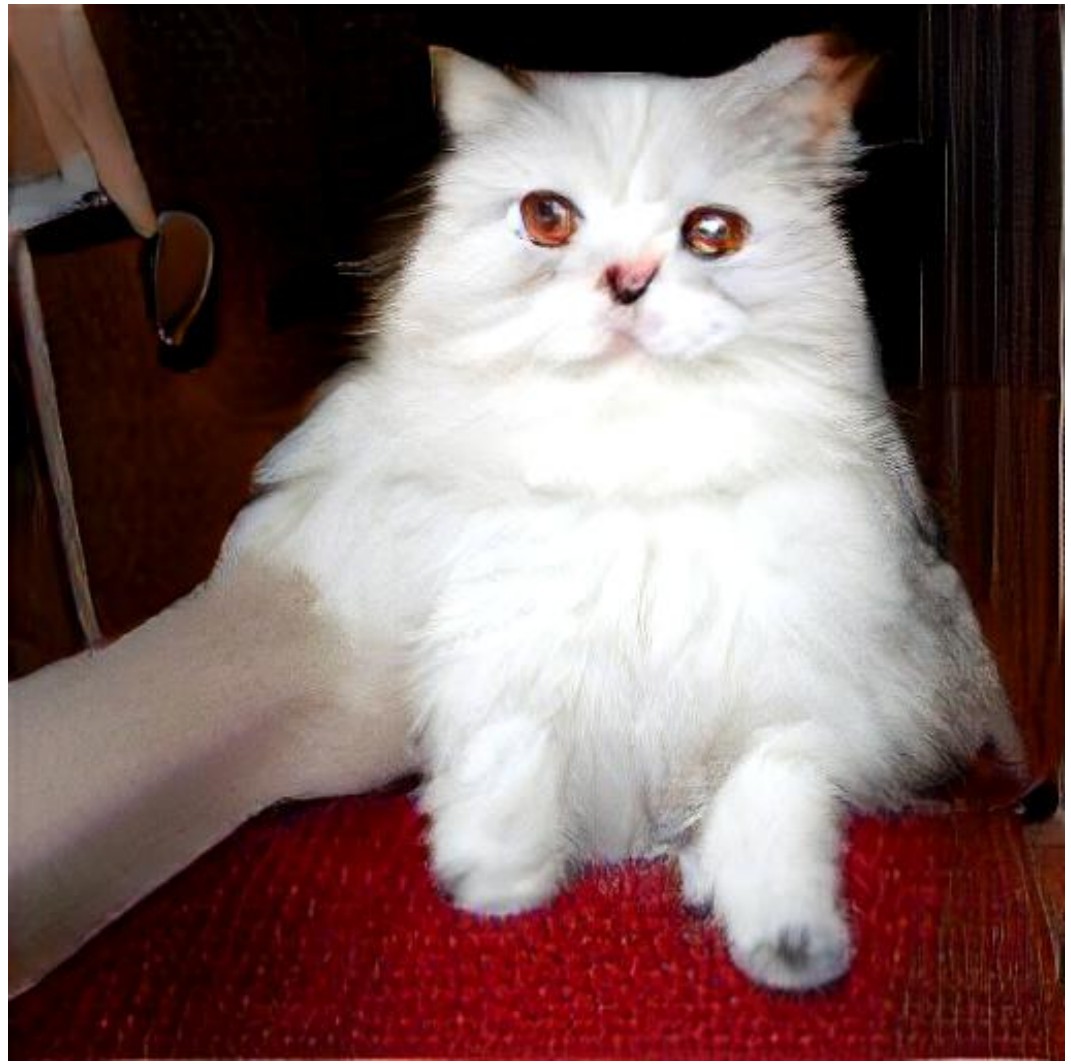

Figure 18: Full $512 \times 512$ resolution sample from NSB-GAN-W with pre-trained sampler.

| Sampler | Decoder | Resolution | FID | IS |
|---|---|---|---|---|
| **Pretrained-256-64** | ESRGAN-W | 256 | 13.53 | 3.182 |
| **Pretrained-256-64** | ESRGAN-W (Church) | 256 | 7.886 | 2.784 |

Table 10: The above pre-trained samplers are StyleGAN-2 model trained on LSUN Church dataset. Even our decoder trained only on ImageNet reaches a competitive FID of 13.53, but evidently can be improved by training a new decoder on LSUN Church dataset. These results show that our NSB-GAN models can work with different samplers than BigGAN and that our decoder generalizes fairly well to unseen dataset. Therefore, we do not go about proving the compute efficiency in this case.

## F    SLICING

Figure 32b, shows the difference between the slicing operations of SPN and NSB-GAN. SPN slices the images across the spatial dimensions thus introducing long-term dependencies. In contrast, NSB-GAN decomposes the image along different frequency bands that preserves the global structure of the image in each of the patches.

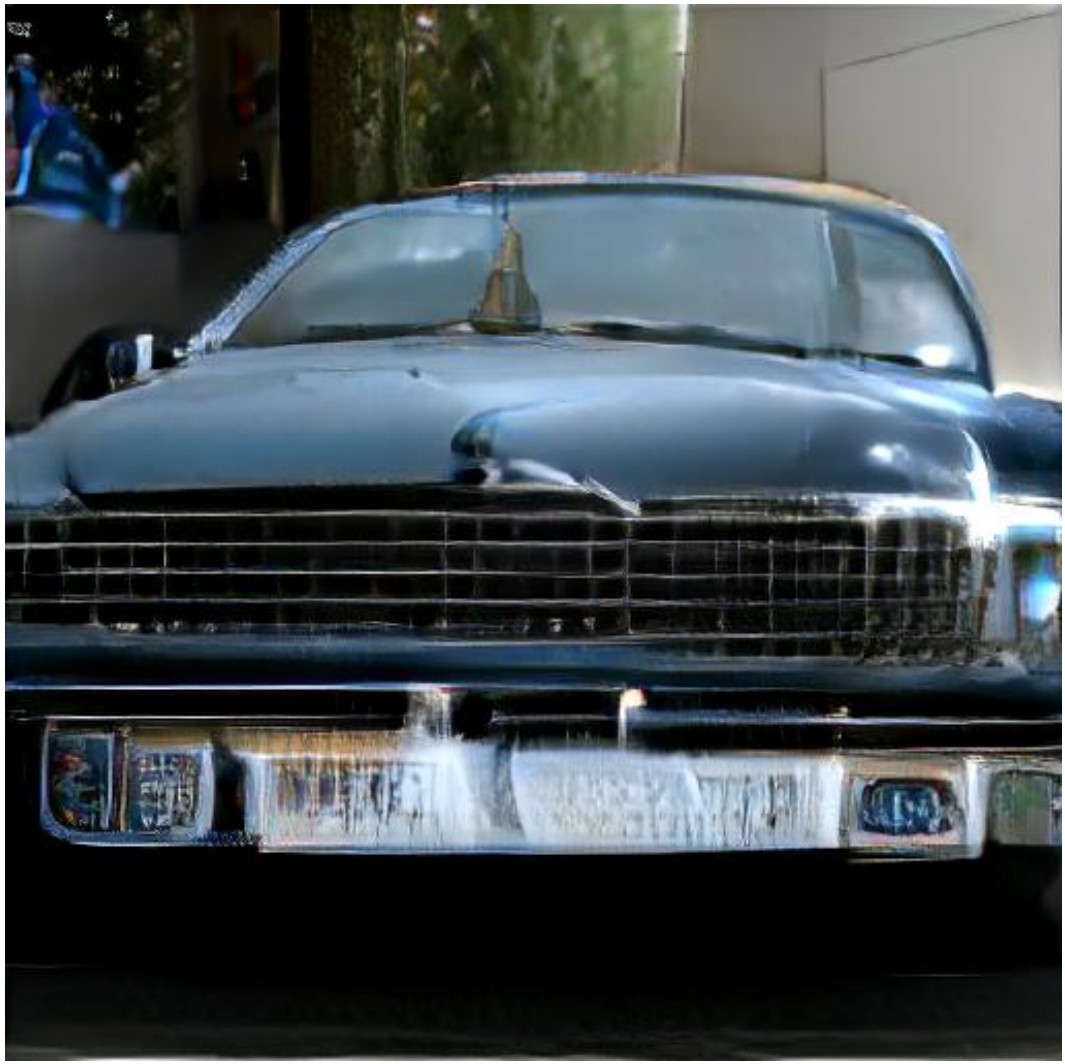

Figure 19: Full $512 \times 512$ resolution sample from NSB-GAN-W with pre-trained sampler.

## G    WAVELET TRANSFORM.

Wavelet transformation of an image (illustrated in Figure 1) is a two-step recursive process that splits the original image into four equal-sized patches, each a representation of the entire image in different frequency bands. In the first step, a low-pass filter (LPF) and a high pass-filter (HPF) are applied to the original image. This produces two patches of the same size as the original image. Since the application of LPF and HPF leads to redundancy, we can apply Shannon-Nyquist theorem to downsample these patches by half without losing any information. In step two, the same process is repeated on the output of step one, splitting the original image into four equally-sized patches (TL, TR, BL and BR). TR, BL and BR contain increasingly higher frequencies of the input image, preserving horizontal, vertical and diagonal edge information, respectively (contributing to the sharpness of the image).

## H    NSB-GAN AT $512 \times 512$

At $512 \times 512$ resolution, our model outperforms the BigGAN model. We believe this to be the case mainly because of how the $512 \times 512$ training data is generated. The ImageNet dataset is natively at $256 \times 256$ (approximately). When training the BigGAN model to generate $512 \times 512$, an interpolation-based method is used to upsample ImageNet images to $512 \times 512$, resulting in noise

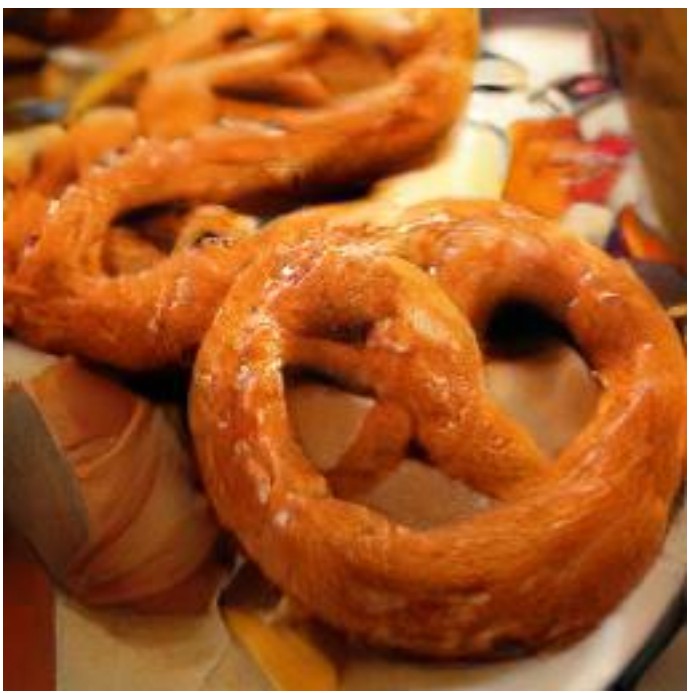

Figure 20: Full $256 \times 256$ resolution sample from NSB-GAN-P with pre-trained sampler.

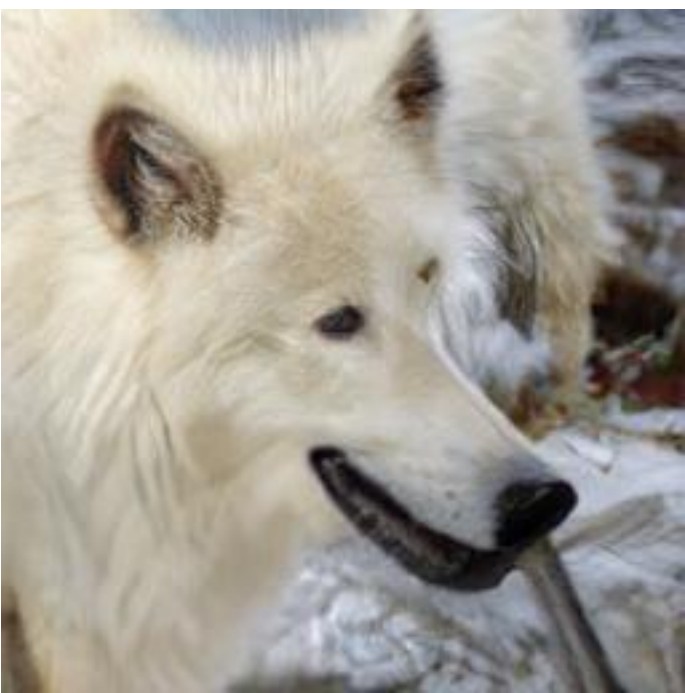

Figure 21: Full $256 \times 256$ resolution sample from NSB-GAN-P with pre-trained sampler.

and blurriness in the upsampled images. In comparison, our method takes samples at $128 \times 128$ and upsamples them using a learned SISR model. This leads to substantially sharper images and therefore better FID scores. In fact, based on our study, it is better to use our model over an end-to-end DGM when learning to generate samples beyond the native resolution of the dataset. To clearly demonstrate this difference, we show samples from the bilinearly interpolated $512 \times 512$ ImageNet

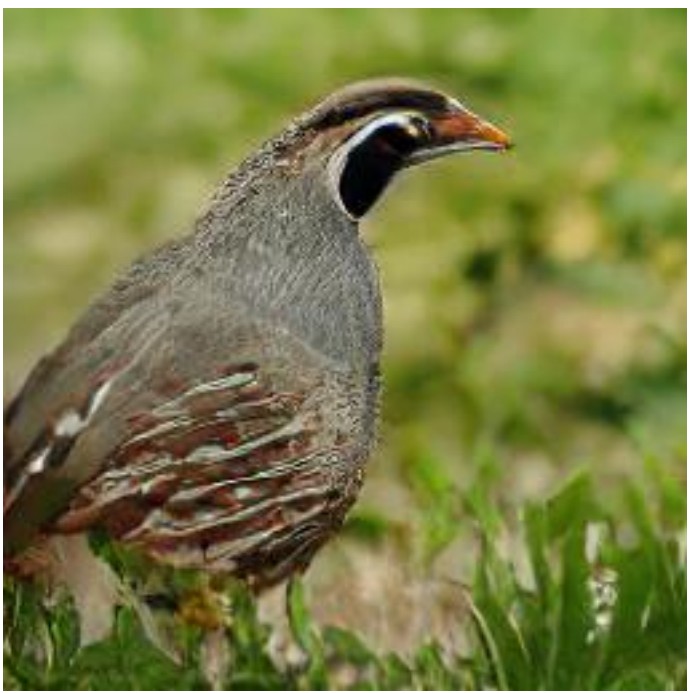

Figure 22: Full $256 \times 256$ resolution sample from NSB-GAN-P with pre-trained sampler.

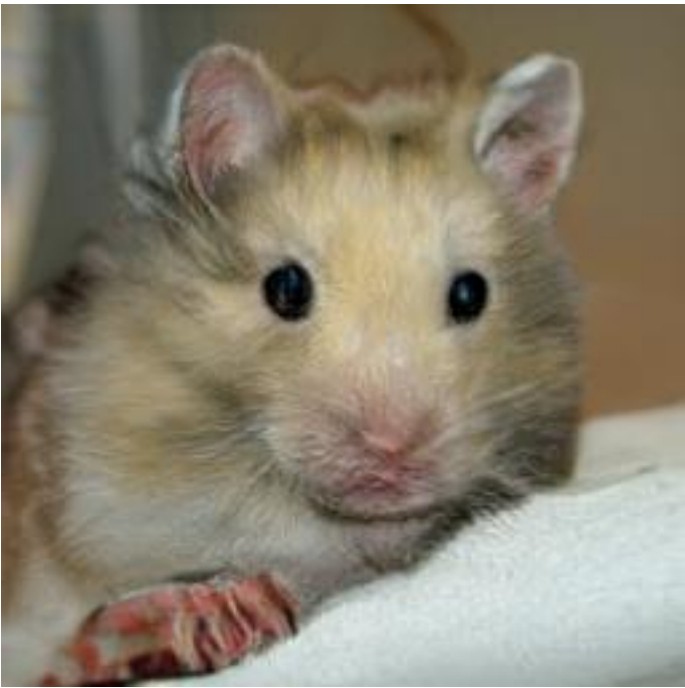

Figure 23: Full $256 \times 256$ resolution sample from NSB-GAN-P with pre-trained sampler.

data and super-resolved version of the same samples with our decoder in Figure 31. Clearly, interpolated images are blurrier than super-resolved images. Since BigGAN is trained to generate this blurry data, compared to our NSB-GAN approach, it performs sub-optimally.

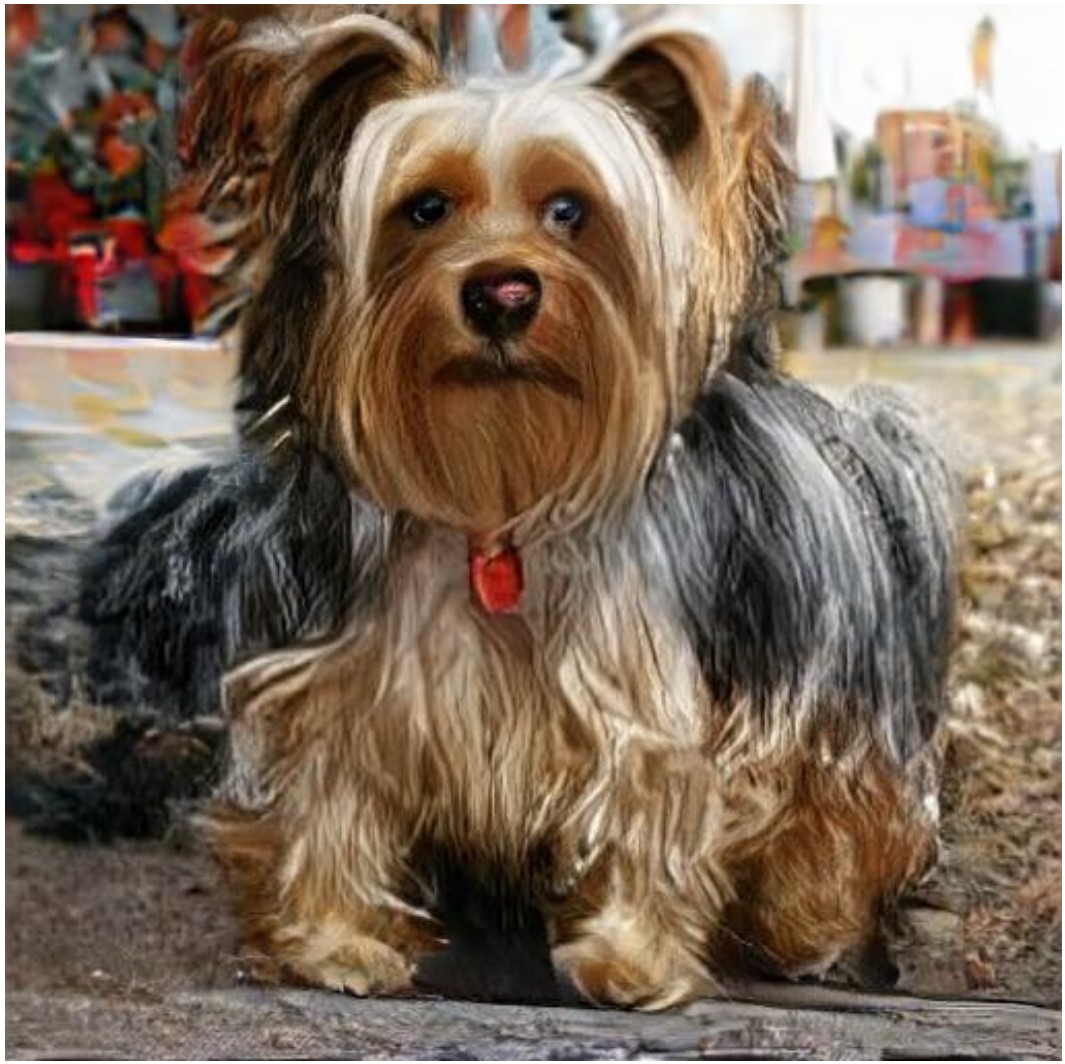

Figure 24: Full $512 \times 512$ resolution sample from NSB-GAN-P with pre-trained sampler.

NSB-GAN beats the BigGAN baseline model on FID at $512 \times 512$ resolution. We hypothesize that this difference in performance is due to the fact that ImageNet dataset at $512 \times 512$ is generated by using pixel-based interpolation up-sampling of the original data.

## I   OFFICIALLY PRE-TRAINED ESRGAN

Applying an officially pre-trained ESRGAN does not perform as well as the ESRGAN-P and -W models. Specifically at $512 \times 512$ resolution, it suffers a slight increase in FID (10.73) and a substantial decrease ($>$100) in IS (52.25).

## J   SAMPLING TIME

Sampling from NSB-GAN takes 0.039 s / image, whereas sampling from the BigGAN model takes 0.029 s / image, at $512 \times 512$ resolution. Therefore, the overhead of upsampling (ESRGAN) is relatively small (0.010 s / image).

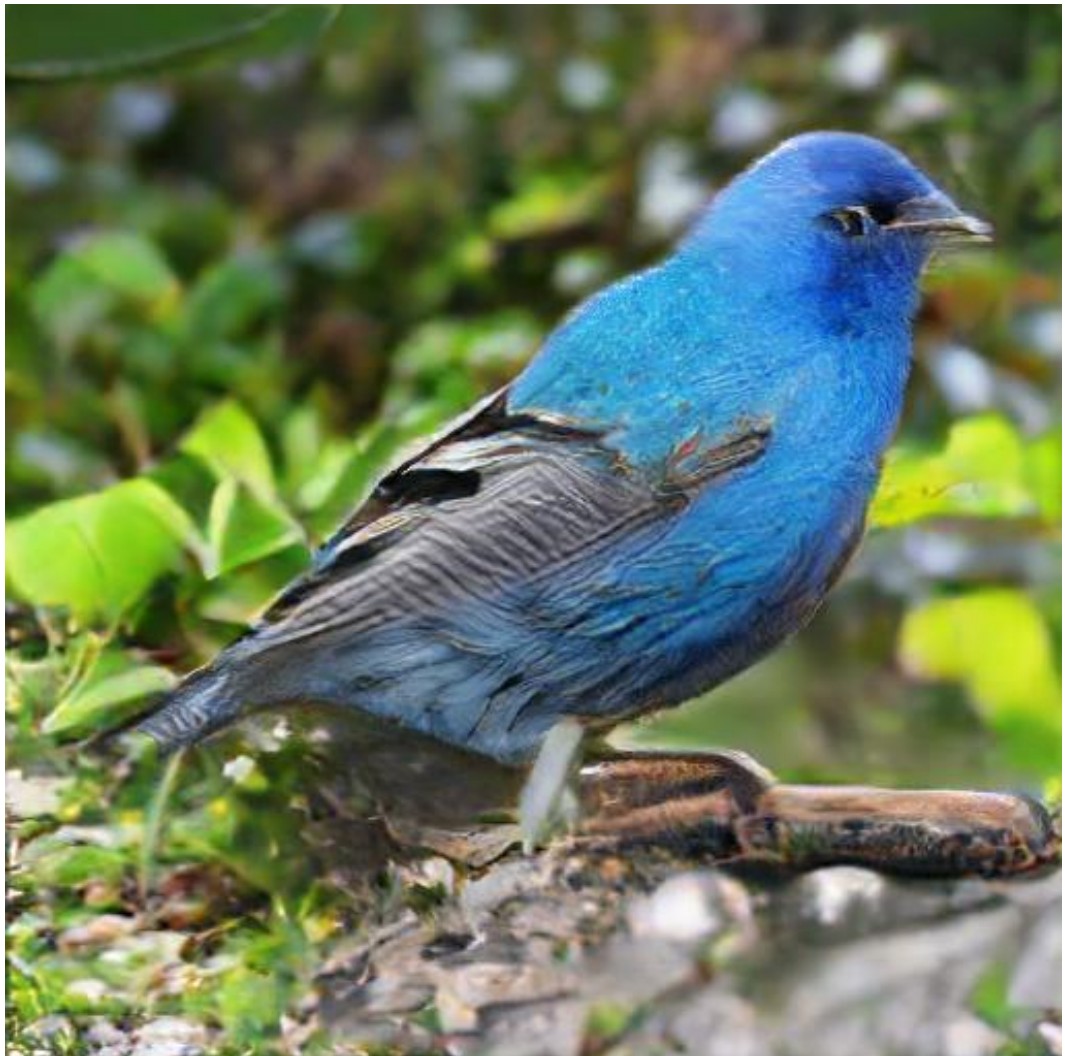

Figure 25: Full $512 \times 512$ resolution sample from NSB-GAN-P with pre-trained sampler.

## K    FINETUNING NSB-GAN-W USING PRE-TRAINED BIGGAN

Since the decoders are trained with real samples from the ImageNet dataset and tested on generated samples from our BigGAN models (learned and pre-trained), our up-sampled outputs could have visual artifacts due to the distribution mismatch. Therefore, we tried finetuning NSB-GAN-W end-to-end with pre-trained BigGAN samples at $256 \times 256$ resolution. We found end-to-end finetuning to increase high-frequency artifacts in the up-sampled images and showcase a few examples in Figure 32. Finetuning also results in worse FID and IS scores of 13.73 and 44.02, respectively. Note that finetuning our model end-to-end requires ground truth samples from an already trained BigGAN at the target resolution. This somewhat defeats the purpose of our approach, which tries to avoid training a BigGAN at the target resolution to begin with.

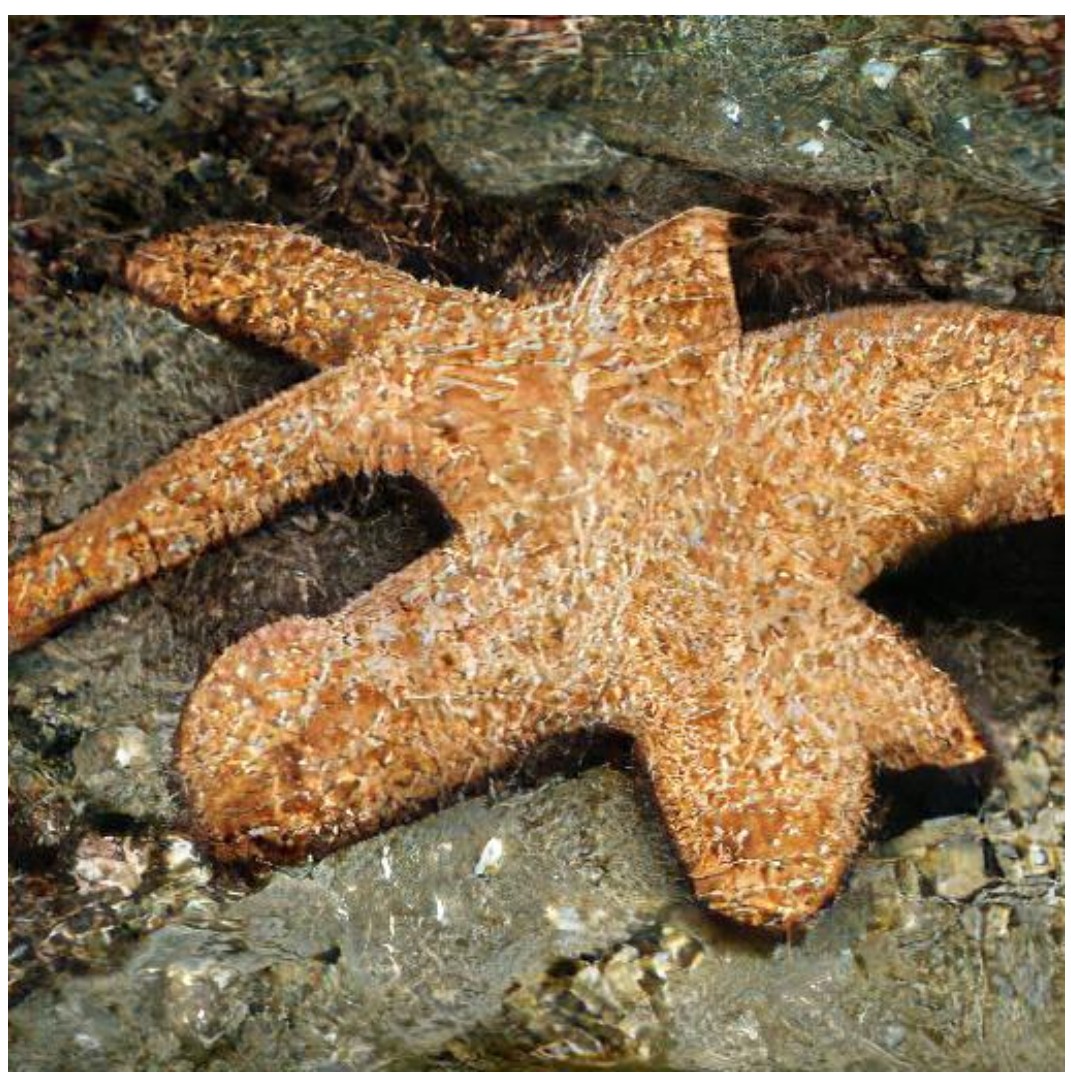

Figure 26: Full $512 \times 512$ resolution sample from NSB-GAN-P with pre-trained sampler.

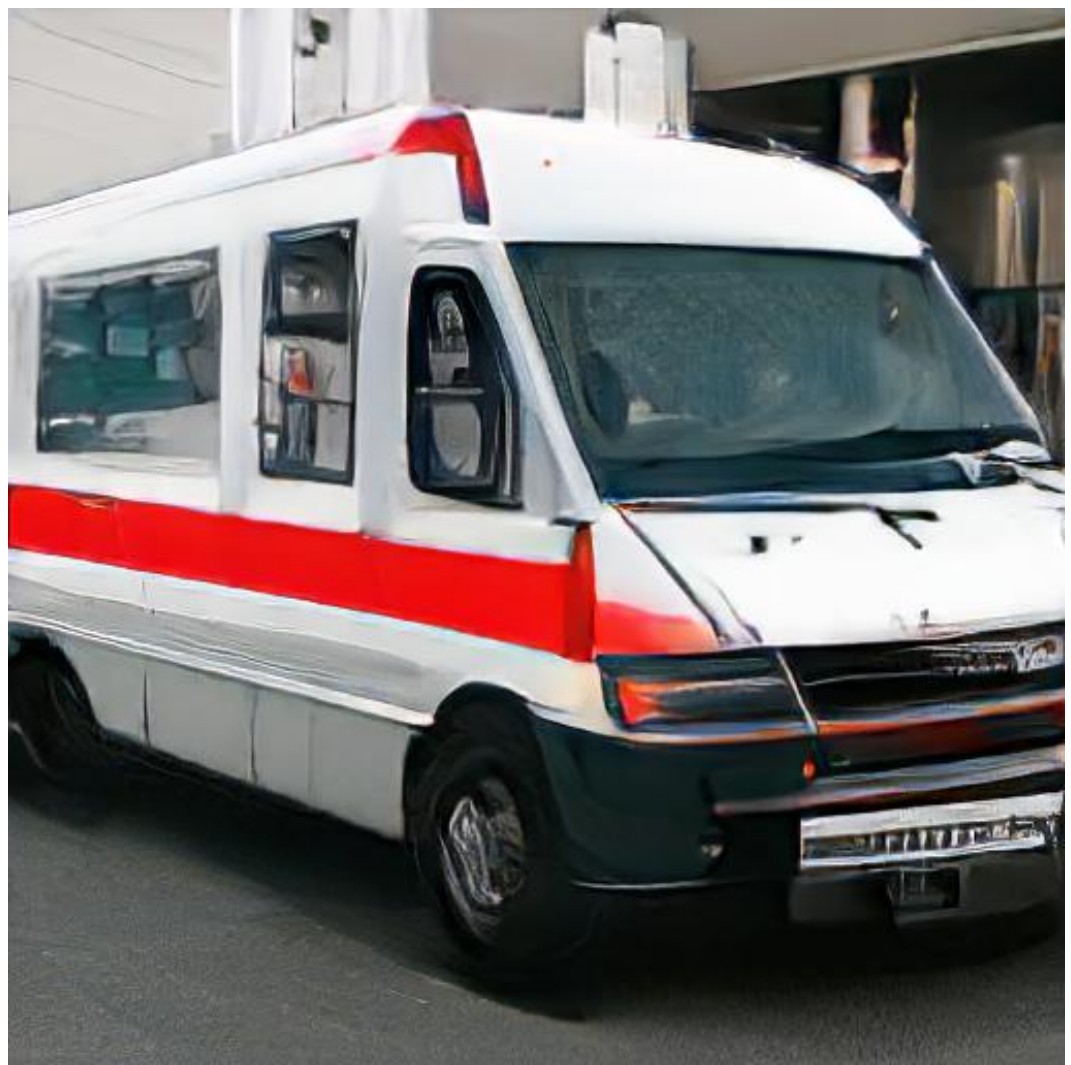

Figure 27: Full $512 \times 512$ resolution sample from NSB-GAN-P with pre-trained sampler.

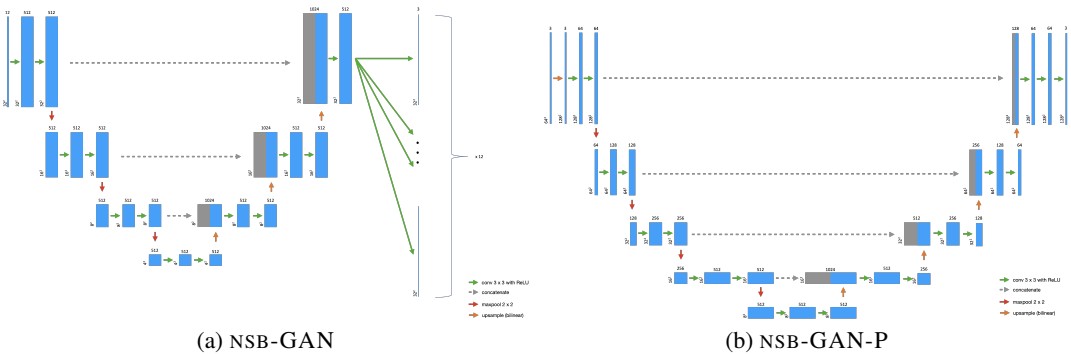

(a) NSB-GAN          (b) NSB-GAN-P

Figure 28: UNet-based decoding architecture for NSB-GAN and NSB-GAN-P models.

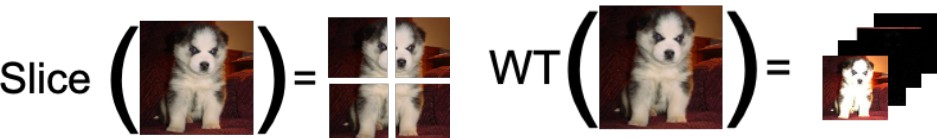

(a) SPN Slicing Operation         (b) NSB-GAN Slicing Operation

Figure 29: Unlike SPN, NSB-GAN slices the images in the frequency domain. As a result each patch contains the entire global structure of the input image. This helps alleviate any long-term dependency issues.

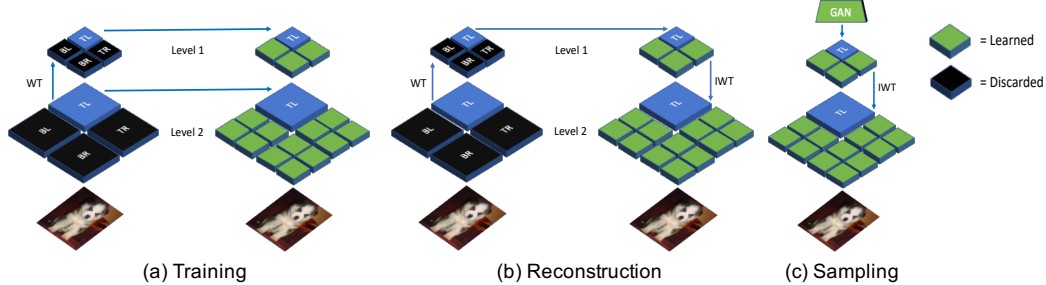

(a) Training        (b) Reconstruction        (c) Sampling

Figure 30: NSB-GAN schematic for (a) training, (b) reconstruction and (c) sampling.

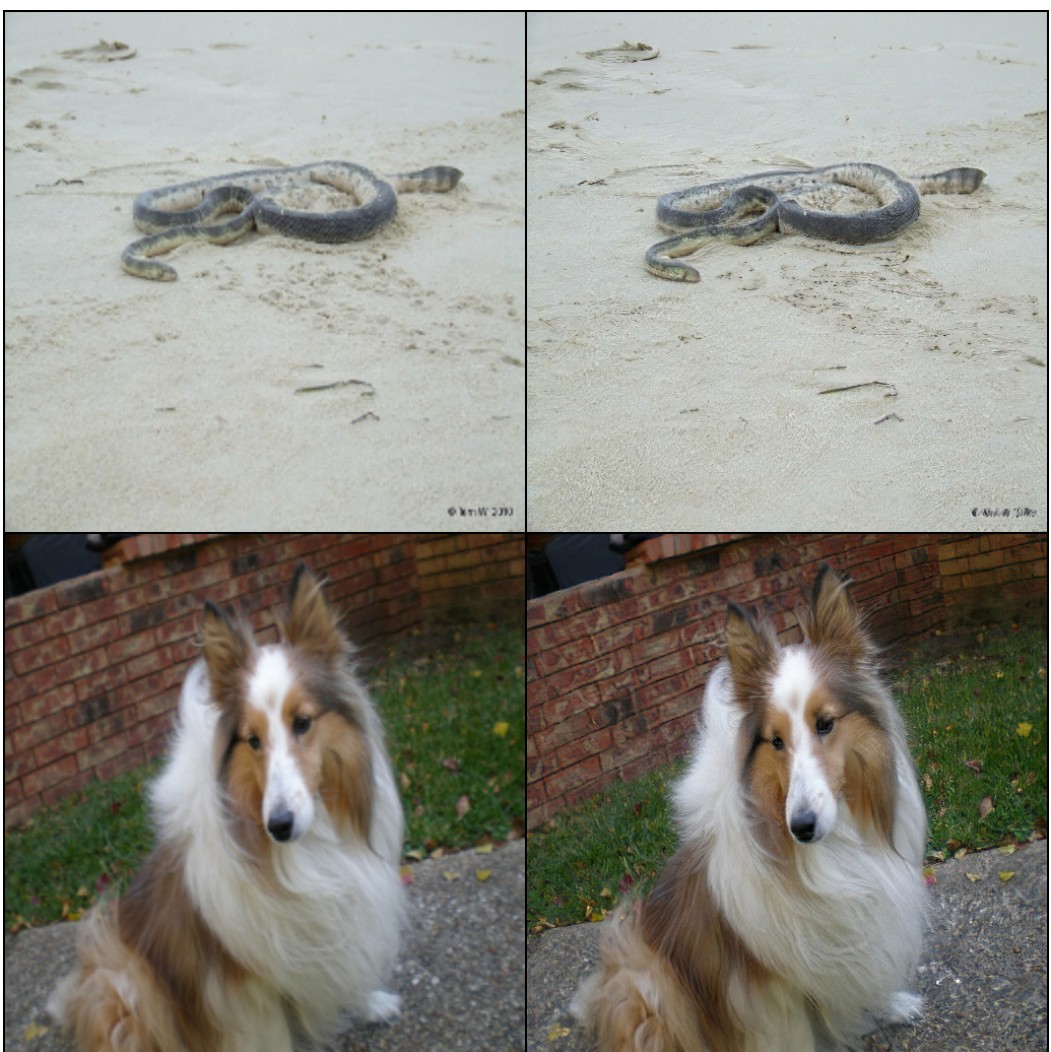

Figure 31: The left column consists of images from the ImageNet dataset at $512 \times 512$ and the right column consists of super-resolved version of the same images using our decoder (from $128 \times 128$ to $512 \times 512$. Clearly, the ImageNet dataset at $512 \times 512$ is blurrier than super-resolved images.

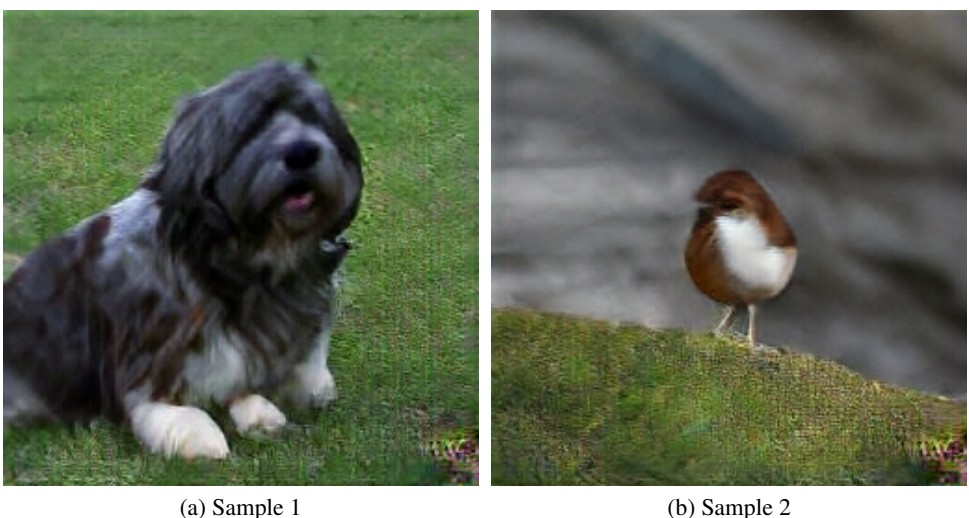

(a) Sample 1                                          (b) Sample 2

Figure 32: Generated samples at $256 \times 256$ after end-to-end finetuning of NSB-GAN-W with pre-trained BigGAN-256. After finetuning, high frequency noise (e.g., checkerboard-like patterns) can be observed in the images.

