# OpenReview forum: "not-so-big-GAN: Generating High-Fidelity Images on Small Compute with Wavelet-based Super-Resolution"
_ICLR.cc/2021/Conference — Reject_

### Official Review · AnonReviewer3 · 2020-10-28
**A cost-effective two-step GAN framework**

**Rating:** 5
**Confidence:** 3

**Review:**

**Summary**

This paper proposes a cost-effective two-step training GAN framework (NSB-GAN). NSB-GAN contains a learned sampler in the wavelet domain, and a decoder to super-resolve images from the wavelet domain to the RGB space. Compared with the baseline BigGAN model their method reduces the cost of training and generates higher quality images at 512x512 resolution.

**Strengths**
- Contributions clearly stated and validated and the paper is clearly written and easy to understand.
- Comprehensive experiments to show the effectiveness of their method.
- Cost-effective than BigGAN and performance is good.

**Weaknesses**
- The idea of this work is not so novel. It seems low-resolution sampler + super-resolution decoder which is a straightforward idea and lacks novelty. And the two parts are trained independently. Why the authors didn't finetune the sampler and decoder networks together in the end so that to get a better sampler from a more correct sampling space.
- For (x4) super-resolution networks, e.g., EDSRGAN, they often fail to generate a high-quality HR image with rich details when the input LR image is low quality. Therefore, what if the LR images from the learned sampler contain do not contains any desired patterns/textures?
- For the NSB-GAN sampler, the authors use a batch size of 512, which is much smaller than that of BigGAN (2048). The authors should give an ablation study to analyze the effect of batch size on the proposed framework.
- From Figure 2 in Appendix A, it seems that wavelet-based downsampling loses more structural information than pixel-based downsampling, which is not consistent with the description in the paper and violates the motivation of wavelet-based super-resolution as well. Please check it.
- I can not understand the claim that "NSB-GAN models reduce the training compute budget by up to four times" in the caption of Table 2. Because Table 2 shows the compute of NSB-GAN is half of BigGAN, not 1/4.
- I notice that NSB-GAN outperforms BigGAN at 512 resolution in terms of min FID, but it shows worse performance at 256 resolution. Which leads to these results? The authors are suggested to give some analysis on this problem.

**Post rebuttal**
I appreciate that the authors answer my questions. After reading through the rebuttal and other reviews,  I partly agree with R1's comments and I would like to downgrade my score by 1. My main concern is the novelty of this method. I disagree that the decoupling of generation from upsampling is interesting, it seems more like an engineering problem. Besides, I found that the authors changed the images in Figure 2 in Appendix A, not only changed the order, which is not consistent with their explanation in rebuttal.

---

> ### Author Response · Authors · 2020-11-15
> **Response**
>
> We thank R3 for the feedback. Here are our responses to their questions in order:
> >The idea of this work is not so novel. It seems low-resolution sampler + super-resolution decoder which is a straightforward idea and lacks novelty.
>
> We respectfully disagree with R3. To the best of our knowledge, the decoupling of generation from upsampling in deep generative models is novel and has not been studied before. Furthermore, our method is the first successful attempt at wavelet space training of DGMs and SISRs. Most importantly, our method is the most energy efficient and the fastest way of training the BigGAN model without any modelling changes. In fact, our method even outperforms the BigGAN model when generating samples beyond the original resolution of the dataset (at 512x512).
>
> The simplicity and straightforwardness of our method is in its implementation. But, as R4 points out, this is in fact a strength; our two-step training framework is very flexible and can be realized using a number of different existing models.
>
> >Why the authors didn't finetune the sampler and decoder networks together in the end...
>
> Finetuning the two models together requires ground truth samples from an already trained BigGAN at the target resolution. This defeats the purpose of our approach, which tries to avoid training a BigGAN at the target resolution to begin with. Following your suggestion, we ran new experiments where we used samples from a pretrained BigGAN at 512x512 to refine our end-to-end model. We found that this leads to high-frequency artifacts in the generated images and worse results on FID/IS (FID 13.73, IS 44.02). We have added these results to Appendix K.
>
> >...what if the LR images from the learned sampler contain do not contains any desired patterns/textures
>
> R3 is correct. We explain this in Section 4.1.1 using Figure 2 from the Appendix.  When the LR resolution falls below 64, the FID suffers an increase due to loss in structural information. We are a bit unclear exactly what weakness this pertains to. Could R3 please clarify?
>
> >For the NSB-GAN sampler, the authors use a batch size of 512, which is much smaller than that of BigGAN (2048). The authors should give an ablation study to analyze the effect of batch size...
>
> Based on the original BigGAN paper, where they do study the impact of batch size, we do think our 64 x 64 model will benefit from a higher batch size. Unfortunately, running this experiment is not possible for two reasons.
> 1. The public implementation of BigGAN does not implement Sync BatchNorm (as we mentioned in the manuscript) which is crucial to see any benefit from higher batch size training.
> 2. We do not have access to a 128-core TPU v3 Pod or equivalently large GPU clusters to run the 2048 mini-batch setting. This is actually one of the main motivations behind our work.
>
>
> >From Figure 2 in Appendix A, it seems that wavelet-based downsampling loses more structural information than pixel-based downsampling...
>
> We sincerely apologize for the confusion. The caption of Figure 2 is incorrect. The rightmost image is indeed wavelet-based downsampling and not the middle column. With the correction, it is evident from the figure that pixel-space downsampling suffers from more information loss and blurriness than wavelet-based downsampling. We have updated the manuscript with the corrected version.
>
> >I cannot understand the claim that "NSB-GAN models reduce the training compute budget by up to four times"
>
> We are referring to the last row of Table 2, where we directly generate samples from the 128x128 pretrained BigGAN sampler and run it through our decoder to generate 512x512 images. We will clarify this in the manuscript.
>
> >I notice that NSB-GAN outperforms BigGAN at 512 resolution in terms of min FID, but it shows worse performance at 256 resolution... The authors are suggested to give some analysis on this problem.
>
> At 512x512, our model outperforms the BigGAN model because of how the 512x512 training data is generated. The ImageNet dataset is natively at 256x256 (approximately). When training the BigGAN model to generate 512x512, an interpolation-based method is used to upsample ImageNet images to 512x512, resulting in gaussian noise in the upsampled images. In comparison, our method takes samples at 128x128 and upsamples them using a learned SISR model. This leads to substantially sharper images and therefore better FID scores. In fact, based on our study, it is better to use our model over an end-to-end DGM when learning to generate samples beyond the native resolution of the dataset. To clearly demonstrate this difference, we have added illustrative samples from the bilinearly interpolated 512x512 ImageNet data and super-resolved version of the same samples with our decoder in Appendix K (Figure 31). Clearly, interpolated images are blurrier than super-resolved images. Since BigGAN is trained to generate this blurry data, compared to our approach, it performs sub-optimally.

---

### Official Review · AnonReviewer4 · 2020-10-29
**compute-efficient deep generative models for large images**

**Rating:** 6
**Confidence:** 4

**Review:**

This paper proposes a new framework NSB-GAN for high-resolution natural image generation with a low computation budget. They first utilize the BigGAN (or StyleGANv2) to generate a low-resolution image, and the wavelet- or pixel-based SR network decodes the intermediate image to the desire resolution. With this simple two-step approach, they successfully train the model on the limited resources where vanilla BigGAN cannot be trained. In addition, NSB-GAN outperforms the BigGAN on high-resolution (>512x512) maybe due to the instability of the BigGAN at extremely high resolution.

Strengths:
+ The motivation of compute-efficient deep generative models for large images is a very important issue, but not many methods have taken this into account. The proposed method tackles this issue in a simple but effective manner.
+ Since the proposed approach does not modify the network architecture, it is flexible, allowing any model compression techniques can be adapted accordingly.

Weaknesses:
- Lack of some critical comparisons: 1) Does the proposed method also outperform the approach that first generates using a pre-trained BigGAN-256 then upsamples using an officially pre-trained ESRGAN? 2) What about the inference time (or complexity) of NSB-GAN compared to BigGAN?
- If the decoders are trained with real samples only (drawn from the imagenet dataset), the upsampled outputs may have visual artifacts due to the mismatched distribution between the train (real) and test (fake gen from sampler) images. For example, the images of the cabinet (Fig3, 4, 5th row) have overly sharpened artifacts that BigGAN does not suffer.

Overall, the suggested work effectively decreases the training time of the BigGAN using a simple idea. However, there are missing comparisons and analyses such as 1) comparison with pre-trained BigGAN -> ESRGAN 2) How the capacity of the SR model affects the FID. And lastly, since the proposed method is pipelining, there are some unexpected artifacts.

---

> ### Author Response · Authors · 2020-11-15
> **Response**
>
> We thank R4 for their feedback and recognising our core contribution. We have run all the three suggested experiments and added the results to the manuscript. We now describe them here.
>
> >Does the proposed method also outperform the approach that first generates using a pre-trained BigGAN-256 then upsamples using an officially pre-trained ESRGAN?
>
> That is correct. Applying an officially pre-trained ESRGAN does not perform as well. Specifically, at 512x512 resolution, it suffers a slight increase in FID (10.73) and a substantial decrease (>100) in IS (52.25). Following your suggestion, we have added this experiment to Appendix I.
>
> >What about the inference time (or complexity) of NSB-GAN compared to BigGAN?
>
> NSB-GAN inference takes 0.039 s / image, whereas BigGAN inference takes 0.029 s / image, at 512x512. Therefore, the overhead of upsampling (ESRGAN) is relatively small (0.010 s / image). We have added these numbers to the Appendix J.
>
> >If the decoders are trained with real samples only (drawn from the imagenet dataset), the upsampled outputs may have visual artifacts due to the mismatched distribution between the train (real) and test (fake gen from sampler) images. For example, the images of the cabinet (Fig3, 4, 5th row) have overly sharpened artifacts that BigGAN does not suffer.
>
> Following R4’s suggestion, we tested finetuning our models, and we found that it increases high-frequency artifacts in the generated images. We have included examples in the manuscript in Appendix K.
> Please note, finetuning the two models together requires ground truth samples from an already trained BigGAN at the target resolution. This somewhat defeats the purpose of our approach, which is trying to avoid training a BigGAN at the target resolution to begin with.

---

### Official Review · AnonReviewer2 · 2020-10-29
**Neat idea, but technical flaws cast doubt on conclusions**

**Rating:** 2
**Confidence:** 3

**Review:**

The paper describes a two-stage generative image model: first, a GAN is trained to output low-resolution images, and another model to then perform single-image superresolution on the results of the first model. The claim is that the resulting model is slightly better than BigGAN-512 using half the compute requirements, in terms of FID. Two variants are described: one that generates in the wavelet decomposition domain (-W), and another that operates in pixel space (-P).

The idea of applying SISR to a GAN output seems potentially novel and useful, but as it stands, I find the paper convoluted and lacking a clear message. I cannot recommend acceptance at this point.

The main issues I see are the following:

1. I suspect much of the image processing is performed incorrectly, casting doubt on the validity of the -P model and rendering the comparison to the wavelet case moot.
2. Results are only reported in numeric form as FID and IS.
3. The argumentation about the properties of the wavelet decomposition seem vague and without clear technical counterparts in the (well-developed) literature on wavelets and image processing.
4. Simultaneous lack of details and overall verbosity; I find it difficult to find the big picture from this paper even after hours of trying.


1. I believe the bilinear downsampling, the basis of the -P variant, is implemented incorrectly. This is visible as clear aliasing in the supplemental Figure 2, rightmost “pixel-space” column. To verify, I extracted the dog and sailboat images from the PDF and applied bilinear downsampling in Matlab – which uses proper pre-filtering before downsampling to remove aliases, unlike for instance the bilinear grid_sample operation in PyTorch – and get a significantly different result; one that does not have the signature aliasing artifacts that remain in the images shown. If, on the other hand, I explicitly turn off antialiasing, the result quite closely matches the right-hand column. To be clear, this a rudimentary mistake in image processing (which is surprisingly prevalent in the ML and vision literature).

This makes me suspect all results of the -P variants are not to be trusted: teaching a GAN to generate aliased images, and then another model to up-res those aliased images, seems like a task that is fundamentally harder than if the aliasing wasn’t there. Hence the worse results are not unexpected.

In particular: I find the conclusions drawn from the results in Table 1 all potentially invalid.  On the other hand, using pretrained samplers, the pixel space versions appear to actually do a little *better* (in terms of FID) than the wavelet ones in Table 2. Comparing the results in the appendix does not appear to reveal large differences.


2. It is well known that metrics like FID and IS do not capture the notion of the quality of a GAN well. They are useful in drawing a picture of how the model performs. While some metrics that better correlate with result quality are known, a satisfactory one hardly exists, so visual inspection and analysis of the results cannot be skipped. I do not approve of pushing them to the appendix.

3. Example: "The functional prior imposed by our deterministic encoder leads to a highly structured representation space made up of low frequency TL patches of images.” What does this mean, precisely? The repeated application of the wavelet approximation coefficient filter followed by decimation by 2 is equivalent to a particular linear downsampling operation applied to the original image; a poor one at that, because the kinds of critically sampled wavelets employed here are known for their aliasing issues (which has long ago led to a preference of using overcomplete bases). Similar language about the “structuredness” of the wavelet representation can be found near Figure 2, where the pixel-space comparison is, I believe, incorrect, as I detail above.

4. What precisely is going on with Equation 2? IWT(…) would appear to be a reconstruction operation that combines the approximation and detail coefficients into an image of 2x the size, in pixel space; then addition of f(W^l_1,1) seems to add hallucinated detail on top. Does f do anything random or is it deterministic? And more pressingly, how is this actually different from the pixel-space version..?

I do not understand the paragraph 3.1.2 “U-Net decoder”. Why “does [it] not take full advantage of the compression that wavelet space modeling brings about”?

---

> ### Author Response · Authors · 2020-11-15
> **Response**
>
> We thank R2 for their in-depth review and would like to address each of the four issues they brought up next.
>
> > I believe the bilinear downsampling, the basis of the -P variant, is implemented incorrectly...
>
> We want to assure R2 that bilinear interpolation (BI) in our model is implemented correctly. Unfortunately, there is a typo in the caption of Figure 2 (Appendix). The image in the rightmost column is created using wavelet-based downsampling and NOT using BI (as incorrectly stated in the caption). So the aliasing you found was in the wavelet-based downsampled image (which is to be expected; your point #3), and not in the bilinearly interpolated image. We sincerely apologize for the confusion and have updated the draft with corrections.
>
> >... teaching a GAN to generate aliased images... seems like a task that is fundamentally harder than if the aliasing wasn’t there. Hence the worse results are not unexpected.
>
> Since the pixel-space downsampling was done using PIL’s implementation of BI, therefore, it does not suffer from aliasing. But please note, even with aliasing, the -W model substantially (by 10 points) outperforms the -P model with our learned decoder, as illustrated in Table 1. In light of this, R2’s claim that the -P model is worse due to aliasing does not hold and our results are not biased.
>
> We hope that these clarifications address your primary concern, re-establishing the validity of our results.
>
> >visual inspection ... cannot be skipped...I do not approve of pushing them to the appendix.
>
> We agree with R2 about FID/IS, and would have loved to include the samples in the main text. However, given the 256x256 and 512x512 sizes of the samples, it would have easily taken several pages of the main text only to show an unsatisfactory number of samples. Therefore, in the appendix, we provide an abundance of qualitative proof -- class-wise samples and full-resolution samples -- comparing the -W and -P models, alongside pretrained BigGAN samples. We request R2 not to penalize us for this, as this is somewhat beyond our control given the resolution of the images.
>
> >The functional prior ... highly structured representation space... What does this mean, precisely?
>
> Here we are not comparing the -W model with the -P model. Instead, we emphasize that, compared to a VAE or VQ-VAE, our latent spaces are very structured (in fact, they are downsampled images) and therefore we are able to use a convolution-based GAN to sample from them.
>
> >the kinds of critically sampled wavelets employed here are known for their aliasing issues
>
> We empirically tested a large number of bases before settling for bior-2.2 as it led to the best result albeit with aliasing. If R2 has any particular suggestions, we would be more than happy to try it.
>
> >What precisely is going on with Equation 2? ... how is this actually different from the pixel-space version..?
>
> Eq 2 illustrates that the decoder first applies an IWT operation to the image (unlike pixel space upsampling) and then uses the learned function f to fill in the missing information. This differs from the pixel-space version in both the up-sampling method and the amount/type of information the function f needs to learn. For example, in Figure 2 (Appendix), structural details are clearly preserved better using WT downsampling as the two downsampling methods (WT and Pixel) result in different amounts and types of information in the LR image.
>
> >I find it difficult to find the big picture from this paper even after hours of trying.
>
> We apologize if R2 found it difficult to gather the big picture of our work. As mentioned in the contributions subsection, NSB-GAN is an energy-efficient method of training large, deep generative models of high-dimensional natural images with orders of magnitude lower training cost. In this framework, we propose two types of models:
>
> 1. -P, that operates in pixel space and works better in higher dimensional setting (because it loses too much information for the SR method to reconstruct in lower dimensional settings)
> 2. -W that operates in the wavelet space and works relatively well in lower dimensional setting than the -P model as WT downsampling results in relatively smaller content loss (Figure 2).
>
> >I do not understand the paragraph 3.1.2 “U-Net decoder”. Why does [it] not take full advantage ...?
>
> Unlike ESRGAN-W, the U-Net-based decoder breaks down an image (e.g., 256x256) into a collection of 32x32 patches by repeatedly applying WT to all TL, TR, BL, and BR quadrants. By doing so, the decoder learns to fill in the details by reconstructing only 32x32 patches and deterministically combines them using IWT. In effect, the decoder skips to ever operate in the full dimensionality of the image. This paradigm can be best understood with Figure 30 (Appendix) and further explanation in Appendix C.3. In contrast, ESRGAN operates on the full dimensionality of the image and does not benefit from the wavelet decomposition as much.

---

### Author Response · Authors · 2020-11-24
**Summary**

Here is a summary of all the new experiments and changes that we made last week in response to the reviewers’ feedback and suggestions. Before the rebuttal period ends tomorrow, it will be very helpful to know if there are any new questions about our work or the new experiments that we ran on your suggestions.

1. We corrected the typo in the caption of Figure 2 in the appendix (that led to the doubts of R2 and R3 about the results).
2. We have added a new experiment to Appendix I as per the suggestion of R4. Here we use a pre-trained BigGAN with a pre-trained ESRGAN and show how it results in sub-optimal performance compared to NSB-GAN.
3. We have added a new experiment to Appendix J as per the suggestion of R4 to compare the inference times for our models that are marginally slower than BigGAN.
4. We added a new experiment to Appendix K as per the suggestion of R3 and R4 to show the impact of end-to-end fine tuning of our model. We do not find it to improve the performance.
5. We have added a clarification to the results in Table 2 as per R3’s question to explain the *up to 4x claim.*

---

### Decision · Program_Chairs · 2021-01-07
**Final Decision**

**Decision:**

Reject

**Comment:**

The reviewers brought up significant concerns that were not resolved by the authors' responses. The concerns are too significant for the paper to be accepted at this time.